# Towards the Explainability of Temporal Graph Networks via Memory Backtracking

## Abstract

Temporal graphs are ubiquitous in real-world applications such as social networks and finance, where Temporal Graph Networks (TGNs) achieve superior predictive accuracy. Understanding which historical events drive specific model predictions enhances trustworthiness of TGNs. Existing explanation methods for TGNs overlook the memory module, the core component that records and updates node histories, leaving unexplored how past events shape memory dynamics and influence the current predictions. To address this challenge, we propose a framework that attributes TGNs predictions through the topology attribution tree and memory backtracking tree. The topology attribution tree captures neighbor influence, including the impact of their memory vectors. Then, we use the memory backtracking tree to quantify how historical events shape memory evolution. Our method satisfies a conservation principle, ensuring that the total contribution of events equals the model's logits. Finally, we introduce optimization objectives to map logits to probabilities. Experiments on seven temporal graph datasets, spanning node property prediction and link prediction tasks, show that our method provides faithful explanations and consistently outperforms four state-of-the-art baselines.

## 1 Introduction

Temporal Graph Networks (TGNs) (Rossi et al., 2020) have gained increasing attention in real-world applications such as social networks (Liu et al., 2025), e-commerce platforms (Yu et al., 2024), and recommendation systems (Tang et al., 2025). TGNs are powerful because they can capture both the graph topology and the temporal evolution of interactions.

A temporal graph can be represented as a sequence of timestamped events (Figure 1 (a)). Each event is represented as $e_k = (v_k, u_k, t_k)$, indicating that source node $v_k$ and destination node $u_k$ have an interaction event at timestamp $t_k$. Each node maintains a memory vector to store its historical state. Each event has its own feature vector. TGNs process these temporal events through two main stages: **memory update** and **embedding generation** (Figure 1 (b)). Events are processed in batches. When an event occurs, its information is received and encoded into a **message** by combining the source and destination node memories with the event feature. Importantly, this message is used to only update the memory of the destination node, and the updated memory is then passed to subsequent batches. In the embedding stage, each node $u_k$ samples its most recent $n$ events from $\mathcal{N}_{u_k}[0, t] = \{e_k = (v_k, u_k, t_k) | \text{the destination node is } u_k\}$, where $t$ is the current time. Then the node computes event embeddings based on node memories and event features, and aggregates them to produce node embeddings. The embeddings are used for current predictions.

Despite their strong predictive performance, TGNs remain black-box models. They offer little transparency into how predictions depend on historical events. Human-understandable explanations are essential for interpreting their decision-making process. Enhancing explainability is critical to improving the trustworthiness of TGNs and ensuring their safe deployment in real-world applications, particularly in high-stakes domains such as fraud detection (Psychoula et al., 2021; Sinanc et al., 2021) and healthcare forecasting (Cutillo et al., 2020; Amann et al., 2020).

Various explanation methods on static graph neural networks have been proposed, such as GNNExplainer (Ying et al., 2019), PGExplainer (Luo et al., 2020), and FlowX (Gui et al., 2023). These methods typically identify a small subset of important edges, nodes, or subgraphs that contribute the most to the prediction. However, these approaches do not directly apply to temporal graphs, as they

ignore **temporal interactions** and fail to capture how the order of events shapes node states. There have been some recent attempts at TGNs explainability, such as T-GNNExplainer (Xia et al., 2022) and TempME (Chen & Ying, 2023). However, these approaches mainly focus on the graph topology, e.g., identifying influential neighbors or high-contribution structural motifs, while neglecting the **temporal evolution of node memories**. Thus, existing temporal explanation methods fail to uncover the **chain of temporal influences** that drives the model's predictions.

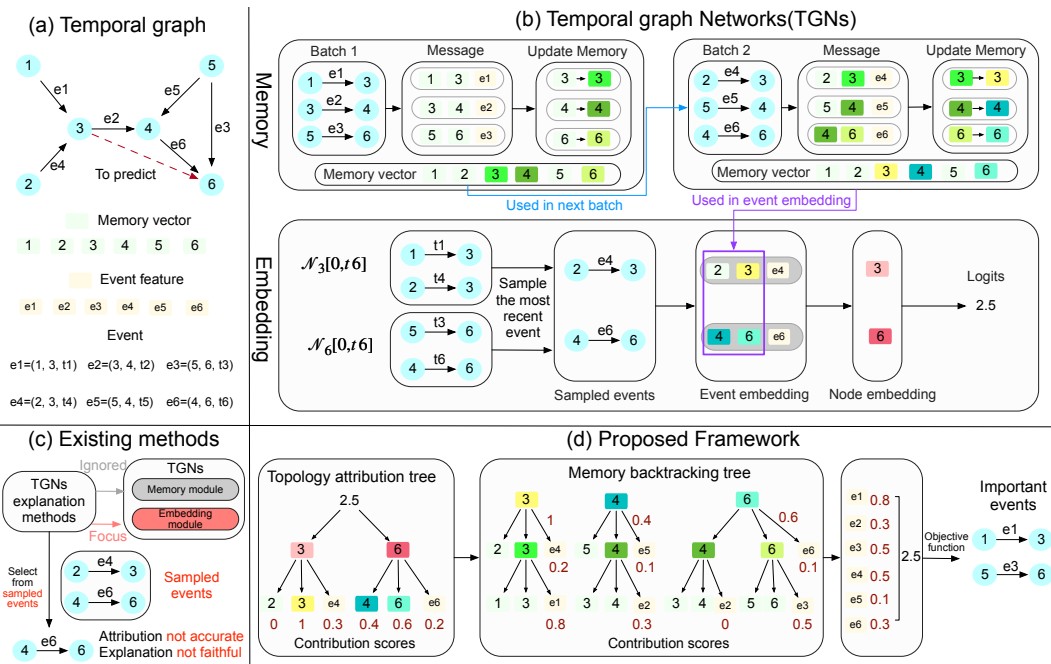

Figure 1: Overview of TGNs and our explainability framework. (a) Input temporal graph, where each node has a memory vector and each event has features. (b) TGNs have two steps: memory update and embedding generation. A message is constructed from the node memories and the event feature, and is used to update the node memory. The updated memories are carried to subsequent batches. Then, nodes sample recent interactions, compute event embeddings based on node memories and event features, and use these embeddings to generate node embeddings for prediction. (c) Existing explanation methods mainly focus on the embedding module and ignore the temporal evolution of node memories, which leads to inaccurate attributions and unfaithful explanations. (d) Our method first attributes logits to recent interactions and node memories via a topology-attribution tree, then propagates memory contributions back to historical events using memory backtracking trees. For example, according to the memory backtracking tree of node 6, the memory vector of node 6 is traced to event $e_6$ and the previous memories of nodes 4 and 6, which are further traced to events $e_2$, $e_3$, and initial memories of nodes 3–6. By summing the event contributions from both steps, we obtain the final contribution of each event, and the total contribution across all events equals the logits. Finally, we design a objective function to select the important events.

To address the challenge of temporal graph explanations, we propose a framework that attributes the predictions of TGNs to historical events by jointly considering the **spatial interactions** among neighbors and the **temporal evolution of node memories** (Figure 1 (d)). From the **spatial** perspective, we introduce a **topology attribution tree** that attributes the logits of TGNs to recent events and their associated node memories. From the **temporal** perspective, we propose a **memory backtracking tree** to propagates memory contributions from the previous step to the historical events responsible for memory updates, recursively tracing how past interactions shape memory evolution. Therefore, it captures how past interactions shape memory evolution. By integrating temporal and spatial attributions, our method guarantees contribution conservation, ensuring that the total attribution of historical events equals the TGNs output logits. To further handle the nonlinear mapping from logits to probabilities, we design an optimization objective that identifies important events without relying on heuristic top-k selection. Experiments on seven temporal graph datasets, spanning both node property prediction and link prediction, demonstrate the effectiveness of our method, which consistently outperforms four state-of-the-art baselines in explaining predicted probabilities.

## 2 RELATED WORK

**GNNs Explainability** Explainability methods for Graph Neural Networks can be broadly categorized into instance-level and model-level approaches (Yuan et al., 2022). Instance-level methods include Gradient/Features-based techniques, such as CAM and GradCAM (Pope et al., 2019), which identify important nodes using gradients. Perturbation-based methods, such as GNNexplainer (Ying et al., 2019), PGExplainer (Luo et al., 2020), learn edge masks by maximizing mutual information to explain the predicted class distribution. Decomposition-based methods, such as GNN-LRP (Schnake et al., 2020), extend the original LRP (Bach et al., 2015) algorithm to GNNs and attribute the importance to graph walks. Surrogate-based methods, like GraphLime (Huang et al., 2020), build a surrogate model with kernel-based feature selection to provide node feature explanations. However, these methods are designed for static graphs and cannot explain temporal graph models. They fail to capture the temporal dependency mixed with the graph topology.

**TGNs Explainability** TGNNExplainer (Xia et al., 2022) is the first explainer tailored for TGNs, which relies on the MCTS algorithm to search for a combination of the explanatory events. TempME (Chen & Ying, 2023) extracts the most interaction-related motifs based on the information bottleneck principle. Recent work (He et al., 2022) utilizes the probabilistic graphical model to generate explanations for discrete time series on the graph, leaving the continuous-time setting underexplored. However, these methods overlook the memory module in TGNs, which is responsible for maintaining and updating node states over time. As a result, they fail to capture how historical interactions accumulate in memory and directly shape future predictions, leaving the core mechanism of TGNs unexplained.

## 3 PRELIMINARIES AND PROBLEM FORMULATION

### 3.1 TEMPORAL GRAPH NETWORKS

A temporal graph is defined as a function of timestamp $t$, denoted by $\mathcal{G}(t) = \{\mathcal{V}(t), \mathcal{E}(t)\}$, where $\mathcal{V}(t)$ and $\mathcal{E}(t)$ represent the set of nodes and events observed before timestamp $t$. Each event $e_k = (v_k, u_k, t_k) \in \mathcal{E}(t)$ indicates an interaction between source node $v_k$ and destination node $u_k$ at timestamp $t_k$, where $t_k < t$. Let $\mathbf{x}_{u_k} \in \mathbb{R}^{d_v}$ and $\mathbf{x}_{e_k} \in \mathbb{R}^{d_e}$ denote feature vectors for node $u_k$ and event $e_k$, respectively. In TGNs, each node $u_k$ maintains a memory vector $\mathbf{s}_{u_k}^t \in \mathbb{R}^{d_m}$ at timestamp $t$. $\mathbf{s}_{u_k}^{t^-} \in \mathbb{R}^{d_m}$ denotes the memory vector of node $u_k$ before timestamp $t$.

Temporal Graph Networks (Rossi et al., 2020) can be viewed as an encoder-decoder framework. The encoder maps the temporal graph $\mathcal{G}(t)$ into time-aware node embeddings, while the decoder takes one or more embeddings to obtain task-specific predictions, such as node property prediction or link prediction. The memory of node is updated whenever the node participates in an event. TGNs compute node embedding $\mathbf{z}_{u_k}^t \in \mathbb{R}^{d_z}$ in four steps:

$$\text{Message Function:} \quad \mathbf{m}_{u_k}^t = f_{\text{message}}(\mathbf{s}_{v_k}^{t^-}, \mathbf{s}_{u_k}^{t^-}, t - t_k, \mathbf{x}_{e_k}), e_k = (v_k, u_k, t_k) \tag{1}$$

$$\text{Message Aggregation:} \quad \bar{\mathbf{m}}_{u_k}^t = f_{\text{agg}}(\mathbf{m}_{u_k}^{t_1}, \cdots, \mathbf{m}_{u_k}^{t_b}) \tag{2}$$

$$\text{Memory Updater:} \quad \mathbf{s}_{u_k}^t = f_{\text{update}}(\bar{\mathbf{m}}_{u_k}^t, \mathbf{s}_{u_k}^{t^-}) \tag{3}$$

$$\text{Embedding:} \quad \mathbf{z}_{u_k}^t = \sum_{e_k = (v_k, u_k, t_k) \in \mathcal{N}_{u_k}^n([0,t])} f_{\text{emb}}\left(\mathbf{s}_{u_k}^t, \mathbf{s}_{v_k}^t, \mathbf{x}_{e_k}, \mathbf{x}_{u_k}, \mathbf{x}_{v_k}\right) \tag{4}$$

Here, the **message function** is triggered by an interaction even $e_k = (v_k, u_k, t_k)$ and computes the message for node $u_k$ based on the memory $\mathbf{s}_{u_k}^{t^-}$, $\mathbf{s}_{v_k}^{t^-}$, the event features $\mathbf{x}_{e_k}$, and the elapsed time $t - t_k$. $f_{\text{message}}$ can be either identity map or an MLP. If multiple messages are received in a batch, the **message aggregator** $f_{\text{agg}}$ combines these messages, either by selecting the most recent message or by averaging, and $t_1, \cdots, t_b \leq t$. The **memory updater** updates the node's memory, typically using an LSTM or GRU. let $\mathcal{N}_{u_k}[0, t] = \{e_k = (v_k, u_k, t_k) | \text{the destination node is } u_k\}$ and $\mathcal{N}_{u_k}^n[0, t]$ denote the set of the most recent $n$ interactions from $\mathcal{N}_{u_k}[0, t]$, where $t$ is the current time. For each event, the **embedding** function $f_{\text{emb}}$ combines the memory states of the target node and its neighbor, along with the associated event and node features to produce an event-specific representation. $f_{\text{emb}}$ can be implemented as a temporal graph attention or a graph sum function.

The node embeddings are fed into a MLP for task-specific predictions. For link prediction, given two nodes $v_k$ and $u_k$ at time $t$, the prediction score is obtained as $\hat{y}_{v_k,u_k} = \sigma(f_{\text{mlp}}([\mathbf{z}_{v_k}^t || \mathbf{z}_{u_k}^t]))$, where $[\cdot \| \cdot]$ denotes vector concatenation, $f_{\text{mlp}}$ is a multi-layer perceptron, and $\sigma$ is the sigmoid function. For node property prediction, the embedding of a node $u_k$ is mapped to its label distribution as : $\hat{y}_{u_k} = \text{softmax}(f_{\text{mlp}}([\mathbf{z}_{u_k}^t]))$.

## 3.2 Problem Formulation

Given a sequence of events and a well-trained temporal graph model, a temporal explainer seeks to explain why the model predicts that an event $e_k$ will occur (link prediction) or why it assigns a specific property to node $u_k$ (node property prediction). In particular, the goal is to identify a subset of historical events (explanations) from $\mathcal{E}(t)$ that trigger the model's prediction.

## 4 Method

### 4.1 Overview

The pipeline of our method is shown in Figure 1. Taking link prediction as an example, given a temporal graph and a prediction between nodes $u_k$ and $v_k$ at time $t$, we proceed in three steps. Firstly, according to the topology attribution tree, we attribute the logits to the historical events in $\mathcal{N}_{u_k}^n([0,t])$ and to the memory vectors $s_{u_k}^t, s_{v_k}^t$ involved in these events (Section 4.2). Secondly, we further decompose the contributions of memory vectors through a memory backtracking process, revealing which past interactions most strongly influence these vectors (Section 4.3). Finally, we derive the KL divergence and design objective functions to select the most important events that explain the model prediction (Section 4.4). For example, in Figure 1 (d), $\mathcal{N}_3^1([0,t_6]) = e_4 = (2,3,t_4), \mathcal{N}_6^1([0,t_6]) = e_6 = (4,6,t_6)$. The logits 2.5 are attributed to events $e_4$ and $e_6$, as well as the memory vectors of nodes 2, 3, 4, and 6 via the topology attribution tree. The contribution of node 6's memory vector is 0.6, updated by events $e_3$ and $e_6$. We then construct memory backtracking trees to assign contributions to $e_3$ and $e_6$. Similarly, the contributions of the memory vectors of nodes 2, 3, and 6 are attributed to the historical events that updated their memories. The sum of all contributions is equal as the logits. Finally, we derive the KL divergence and design objective functions to select the most important events that explain the TGNs prediction.

### 4.2 Spatial Decomposition via topology arrtibution tree

We construct the topology attribute tree and apply LRP (Bach et al., 2015) on the Eq. (4) to decompose the target node embedding $\mathbf{z}_{u_k}^t$ into contributions from the target memory $\mathbf{s}_{u_k}^t$, the neighbor memory $\mathbf{s}_{v_k}^t$, and the feature vectors $\mathbf{x}_{u_k}, \mathbf{x}_{v_k}$, and $\mathbf{x}_{e_k}$. Next, we introduce the original LRP method.

**Original LRP method**. LRP attributes the prediction score to the input neurons. Let the activation of a neuron at layer $l+1$ be $h^{l+1} \in \mathbb{R}$, computed as $h^{l+1} = f([h_1^l, \ldots, h_n^l])$, where $f$ may be a linear function or a composition of a linear function with a nonlinear activation such as ReLU. Given the relevance $R^{l+1} \in \mathbb{R}$, LRP distributes it $h^{l+1}$ to the inputs according to the following equation and satisfy the conservation property: $\sum_i R_i^l = R^{l+1}$.

$$R_i^l = \frac{h_i^l w_i^l}{\sum_{i'} h_{i'}^l w_{i'}^l} R^{l+1},$$

(5)

where $w_i^l$ is connection weight from the input neuron $h_i^l$ to the neuron $h^{l+1}$.

For a multi-layer network, LRP redistributes the relevance score from the output layer back to the input layer by recursively applying the Eq. (5). Let the final prediction logit be assigned as relevance score $R^L$, where $L$ denotes the output layer. The relevance of an input neuron $h_i^0$ is then obtained by successively propagating through all intermediate layers, while preserving the conservation property: $\sum_i R_i^0 = R^L$

$$R_i^0 = \sum_{j,\ldots,k} \frac{h_i^0 w_i^0}{\sum_{i'} h_{i'}^0 w_{i'}^0} \cdot \frac{h_j^1 w^1 j}{\sum_{j'} h_{j'}^1 w_{j'}^1} \cdots \frac{h_k^{L-1} w_k^{L-1}}{\sum_{k'} h_{k'}^{L-1} w_{k'}^{L-1}} \cdot R^{(L)}.$$

(6)

**LRP on** $f_{\text{emb}}$. We apply LRP to $f_{\text{emb}}$. We use the temporal graph sum function as an example to illustrate how LRP attributes the logits of TGNs to $\mathbf{s}_{u_k}^t$, $\mathbf{s}_{v_k}^t$, $\mathbf{x}_{u_k}$, $\mathbf{x}_{v_k}$, and $\mathbf{x}_{e_k}$. In this setting, Eq. (4) becomes:

$$\mathbf{h}_{u_k}^{t,l} = \left(\mathbf{h}_{u_k}^{t,l-1} \| \tilde{\mathbf{h}}_{u_k}^{t,l}\right)\mathbf{W}_2^l, \tilde{\mathbf{h}}_{u_k}^{t,l} = \text{ReLU}(\hat{\mathbf{h}}_{u_k}^{t,l}), \hat{\mathbf{h}}_{u_k}^{t,l} = \sum_{e_k \in \mathcal{N}_{u_k}^n} \left(\mathbf{h}_{v_k}^{t,l-1} \| \mathbf{x}_{e_k} \| \phi(t-t_k)\right)\mathbf{W}_1^l. \quad (7)$$

Where $\mathbf{h}_{u_k}^{t,l} \in \mathbb{R}^{d_m}$ denotes the embedding of node $u_k$ at layer $l$ and time $t$, initialized as $\mathbf{h}_{u_k}^{t,0} = \mathbf{s}_{u_k}^t + \mathbf{x}_{u_k}$. The function $\phi(\cdot)$ is a time encoding. $\mathbf{W}_2^l \in \mathbb{R}^{d_m \times d_m}$ and $\mathbf{W}_1^l \in \mathbb{R}^{d_m \times d_m}$ are parameters, and $(\cdot \| \cdot \| \cdot)$ denotes vector concatenation. The final embedding is $\mathbf{z}_{u_k}^t = \mathbf{h}_{u_k}^{t,L} \in \mathbb{R}^{d_z}$, where $L$ is the number of layers.

Let $\mathbf{A}^l = \mathbf{W}_2^l[1:d_m,:] \in \mathbb{R}^{d_m \times d_m}, \mathbf{B}^l = \mathbf{W}_2^l[d_m+1:\text{end},:] \in \mathbb{R}^{d_m \times d_m}, \mathbf{C}^l = \mathbf{W}_1^l[1:d_m,:] \in \mathbb{R}^{d_m \times d_m}, \mathbf{D}^l = \mathbf{W}_1^l[d_m+1:d_m+d_e,:] \in \mathbb{R}^{d_e \times d_m}, \mathbf{E}^l = \mathbf{W}_2^l[d_m+d_e+1:\text{end},:] \in \mathbb{R}^{1 \times d_m}$ correspond to parameter sub-blocks for self-embeddings, neighbor embeddings, edge features, and temporal encoding. Applying LRP to Eq. (7), and backpropagating through $L$ layers yields:

$$R\left(\mathbf{h}_{u_k}^{t,0} \to \mathbf{z}_{u_k}^t\right) = \text{diag}\left(\mathbf{h}_{u_k}^{t,0}\right)\left(\prod_{l=1}^L \mathbf{A}^l\right), R\left(\mathbf{s}_{u_k}^t \to \mathbf{z}_{u_k}^t\right) = \text{diag}\left(\mathbf{s}_{u_k}^t\right)\left(\prod_{l=1}^L \mathbf{A}^l\right). \quad (8)$$

Where $\text{diag}(\cdot)$ denotes the diagonalization operator that maps a vector to a diagonal matrix. The $R\left(\mathbf{h}_{u_k}^{t,0} \to \mathbf{h}_{u_k}^{t,L}\right)$, $R\left(\mathbf{s}_{u_k}^{t,0} \to \mathbf{h}_{u_k}^{t,L}\right) \in \mathbb{R}^{d_m \times d_z}$ denote the contribution matrices, where each entry quantifies the contributions an input dimension of $\mathbf{h}_{u_k}^{t,0}$ or $\mathbf{s}_{u_k}^t$ to a specific output dimension of $\mathbf{z}_{u_k}^t$.

To obtain the contribution of the sampled events at different layers, we suppose that for a node $p_l$ at layer $l$, there exist a path $(p_l, p_{l+1}, \ldots, p_L)$ to pass information to node $u_k$, where $p_L = u_k$ and $p_l \in \mathcal{N}_{p_{l+1}}^n([0,t])$. The event at layer $l$ for node $p_l$ is $e_l = (p_l, p_{l+1}, t_l)$. We define an operator $\mathbf{T}^l(p_l \to p_{l+1}) = \mathbf{C}^{l+1} \text{diag}\left(\tilde{\mathbf{h}}_{p_{l+1}}^{t,l+1}/\hat{\mathbf{h}}_{p_{l+1}}^{t,l+1}\right) \mathbf{B}^{l+1}$. Let $\mathcal{I}_l = \left\{(\mathbf{h}_{p_l}^{t,l}, \mathbf{I}), (\mathbf{x}_{e_l}, \mathbf{D}^{l+1}), (\phi(t-t_l), \mathbf{E}^{l+1})\right\}$. The relevance of $\mathbf{h}_{p_l}^{t,l}$, $\mathbf{x}_{e_l}$ and $\phi(t-t_l)$ to $\mathbf{h}_{u_k}^{t,L}$ can be expressed in a unified form:

$$R\left(\mathbf{u} \to \mathbf{z}_{u_k}^t\right) = \text{diag}(\mathbf{u})\mathbf{M} \text{diag}\left(\tilde{\mathbf{h}}_{p_{l+1}}^{t,l+1}/\hat{\mathbf{h}}_{p_{l+1}}^{t,l+1}\right)\mathbf{B}^{l+1}\left(\prod_{r=l+1}^{L-1} \mathbf{T}^r(p_r \to p_{r+1})\right) \quad (9)$$

Where $/$ denotes the element-wise division and $(\mathbf{u}, \mathbf{M}) \in \mathcal{I}_l$. The derivation is in Appendix A.2.

**Topology attribution tree**. Based on Eq. (8) and Eq. (9), we can construct the topology attribution tree, and attribute contributions to the events and the memories of nodes, as described in Algorithm 1. Let $\mathcal{T}_{\text{topology}}$ represent the topology attribution tree. As outlined in Algorithm 1, we can compute the contribution matrix $\mathbf{C}_{u_k}^t$, where each row corresponds to an event, and the row vector represents the contribution to $\mathbf{z}_{u_k}^t$. Additionally, we obtain a set of matrices $\{\mathbf{M}_{p \to u_k}^t | p \text{ is a leaf node of } \mathcal{T}_{\text{topology}}\}$. Here, $\mathbf{M}_{p \to u_k}^t \in \mathbb{R}^{d_m \times d_z}$ denotes the contribution of node $p$'s memory vector to $\mathbf{z}_{u_k}^t$. This process follows the conservation property, i.e., $\sum_p \mathbf{1}^\top \mathbf{M}_{p \to u_k}^t + \mathbf{1}^\top \mathbf{C}_{u_k}^t = \mathbf{z}_{u_k}^t$

## 4.3 TEMPORAL DECOMPOSITION VIA MEMORY BACKTRACKING

In Algorithm 1, we obtain a set of the memory matrices $\{\mathbf{M}_{p \to u_k}^t | p \text{ is a leaf node of } \mathcal{T}_{\text{topology}}(u_k, t)\}$ and the event contribution matrix $\mathbf{C}_{u_k}^t$. For each $\mathbf{M}_{p \to u_k}^t$, we construct a memory backing tree to attribute the contributions of node memory to the historical events responsible for updating the target node. In this process, we will update the $\mathbf{C}_{u_k}^t$. For example, in Figure 1 (d), the contribution of the memory vector for node 6 is 0.6. Through the memory backing tree, this value is further decomposed into the contributions of events $e_3$ and $e_6$, with 0.5 and 0.1, respectively. These events correspond to the updates of the node memory. After the memory backtracking process, $\mathbf{1}^\top \mathbf{C}_{u_k}^t = \mathbf{z}_{u_k}^t$. We use $\mathbf{M}_{u_k \to u_k}^t$, the contribution matrix of $\mathbf{s}_{u_k}^t$ to $\mathbf{z}_{u_k}^t$, as an example to demonstrate how to construct the memory backtracking tree and attribute the contributions to the historical events.

**LRP on** $f_{\text{update}}$. We apply the LRP method on Eq. (1)-Eq. (3) to attribute the contributions. The Eq. (1)-Eq. (3) define the memory update module in TGNs. To illustrate the backtracking construction, we set $f_{\text{update}}$ as a GRU, $f_{\text{agg}}$ as the most recent message selector, and $f_{\text{message}}$ as the identity

---

**Algorithm 1** Construction of the topology attribution tree and relevance propagation

---

1: **Input**: target node $u_k$, node embedding $\mathbf{z}_{u_k}^t$, number of layers $L$
2: Initialize $\mathcal{T}_{\text{topology}} \leftarrow \{(u_k, L)\}$ as a tree with root node, set frontier $\leftarrow \{(u_k, L)\}$
3: Initialize contribution matrices: $\mathbf{M}_{p \rightarrow u_k}^t \leftarrow 0$, for all nodes $p$; $\mathbf{C}_{u_k}^t[e_l] \leftarrow 0$ for all events $e$
4: **for** $l = L - 1$ down to $0$ **do**
5:      next $\leftarrow \emptyset$
6:      **for** each parent $(p_{l+1}, l+1) \in$ frontier **do**
7:          **for** each event $e_l = (p_l, p_{l+1}, t_l) \in \mathcal{N}_{p_l}^n[0, t]$ **do**
8:              Insert $(p_l, l)$ and $(\mathbf{x}_{e_l}, l)$ as children of $(p_{l+1}, l+1)$
9:              next $\leftarrow$ next $\cup \{(p_l, l)\}$
10:              Compute $R\left(\mathbf{x}_{e_l} \rightarrow \mathbf{z}_{u_k}^t\right)$, and $R\left(\phi(t - t_l) \rightarrow \mathbf{z}_{u_k}^t\right)$ via Eq. (9)
11:              $\mathbf{C}_{u_k}^t[e_l] \leftarrow \mathbf{C}_{u_k}^t[e_l] + \mathbf{1}^\top R(\mathbf{x}_{e_l} \rightarrow \mathbf{z}_{u_k}^t) + \mathbf{1}^\top R(\phi(t - t_l) \rightarrow \mathbf{z}_{u_k}^t)$
12:          **end for**
13:      **end for**
14:      frontier $\leftarrow$ next
15: **end for**
16: **for** each leaf node $p_0 \in \mathcal{T}_{\text{topology}}$ **do**
17:      Trace the path $\{p_0 \rightarrow p_1 \cdots \rightarrow u_k\}$ and Compute $\hat{\mathbf{R}} = R(\mathbf{h}_{p_0}^{(t,0)} \rightarrow \mathbf{z}_{u_k}^t)$ via Eq. (9)
18:      $\mathbf{M}_{p \rightarrow u_k}^t \leftarrow \mathbf{M}_{p \rightarrow u_k}^t + \text{diag}(\mathbf{s}_{p_0}^t)\,\hat{\mathbf{R}}$, $\mathbf{C}_{u_k}^t[e_0] \leftarrow \mathbf{C}_{u_k}^t[e_0] + \mathbf{1}^\top \text{diag}(\mathbf{x}_{p_0})\,\hat{\mathbf{R}}$
19: **end for**
20: Compute $\hat{\mathbf{R}} = R(\mathbf{h}_{u_k}^{(t,0)} \rightarrow \mathbf{z}_{u_k}^t)$ according to Eq. (8) and $\mathbf{M}_{u_k \rightarrow u_k}^t \leftarrow \mathbf{M}_{u_k \rightarrow u_k}^t + \hat{\mathbf{R}}$
21: **Output:** $\mathcal{T}_{\text{topology}}(u_k, t)$, the contribution matrix $\mathbf{C}_{u_k}^t$, and a set of contribution matrices $\{\mathbf{M}_{p \rightarrow u_k}^t | p \text{ is a leaf node of } \mathcal{T}_{\text{topology}}(u_k, t)\}$

---

map concatenating node memories, time differences, and event features. Eq. (3) can be rewritten as

$$\mathbf{r}_{u_k}^t = \sigma\left(\bar{\mathbf{m}}_{u_k}^t \mathbf{W}_r + \mathbf{s}_{u_k}^{t-} \mathbf{U}_r + \mathbf{b}_r\right), \quad \mathbf{g}_{u_k}^t = \sigma\left(\bar{\mathbf{m}}_{u_k}^t \mathbf{W}_g + \mathbf{s}_{u_k}^{t-} \mathbf{U}_g + \mathbf{b}_g\right),$$

$$\tilde{\mathbf{s}}_{u_k}^t = \tanh\left(\bar{\mathbf{m}}_{u_k}^t \mathbf{W}_h + \mathbf{r}_{u_k}^t \odot (\mathbf{s}_{u_k}^{t-} \mathbf{U}_h) + \mathbf{b}_h\right), \quad \mathbf{s}_{u_k}^t = (1 - \mathbf{g}_{u_k}^t) \odot \mathbf{s}_{u_k}^{t-} + \mathbf{g}_{u_k}^t \odot \tilde{\mathbf{s}}_{u_k}^t.$$

Where, $\mathbf{W}_r, \mathbf{W}_g, \mathbf{W}_h, \mathbf{U}_r, \mathbf{U}_g, \mathbf{U}_h, \mathbf{b}_r, \mathbf{b}_g, \mathbf{b}_h$ are the parameters in the GRU.

Let $\mathbf{F}^t = \mathbf{g}_{u_k}^t \odot \text{diag}\left(\frac{\mathbf{g}_{u_k}^t \odot \tilde{\mathbf{s}}_{u_k}^t}{\mathbf{s}_{u_k}^t}\right)$, $\mathbf{H}^t = \bar{\mathbf{m}}_{u_k}^t \mathbf{W}_h + \mathbf{r}_{u_k}^t \odot (\mathbf{s}_{u_k}^{t-} \mathbf{U}_h)$, using LRP method, the contribution scores of $\mathbf{s}_{u_k}^{t-}$ $\bar{\mathbf{m}}_{u_k}^t$ to $\mathbf{s}_{u_k}^t$ is

$$R\left(\bar{\mathbf{m}}_{u_k}^t \rightarrow \mathbf{s}_{u_k}^t\right) = \text{diag}\left(\bar{\mathbf{m}}_{u_k}^t\right) \mathbf{W}_h \text{diag}\left(\frac{\mathbf{F}^t}{\mathbf{H}^t}\right), \tag{10}$$

$$R\left(\mathbf{s}_{u_k}^{t-} \rightarrow \mathbf{s}_{u_k}^t\right) = \text{diag}\left(\mathbf{s}_{u_k}^{t-}\right) \mathbf{U}_h \ \text{diag}\left(\frac{\mathbf{r}_{u_k}^t \odot \mathbf{F}^t}{\mathbf{H}^t}\right) + \text{diag}\left(\frac{(1 - \mathbf{g}_{u_k}^t) \odot \mathbf{s}_{u_k}^{t-}}{\mathbf{s}_{u_k}^t}\right). \tag{11}$$

Where $-$ denotes the element-wise division. Since $\bar{\mathbf{m}}_{u_k}^t$ is the most recent message (a concatenation of source memory, destination memory, and event features), the decomposition is

$$R\left(\bar{\mathbf{m}}_{u_k}^t \rightarrow \mathbf{s}_{u_k}^t\right) = \left[\ R(\mathbf{s}_{u_k}^{t-} \rightarrow \mathbf{s}_{u_k}^t),\ R(\mathbf{s}_{v_k}^{t-} \rightarrow \mathbf{s}_{u_k}^t),\ R(\mathbf{x}_{e_k} \rightarrow \mathbf{s}_{u_k}^t)\ \right]^\top. \tag{12}$$

**Memory backtracking tree**. To trace node memory, we record which events triggered updates. Let $\mathcal{H}(u_k, t) = \{\ e_k = (v_k, u_k, t_k) \in \mathcal{E}(t)\ :\ t_k < t\ \wedge\ \text{Update}(u_k, t_k)\ \}$, where $\text{Update}(u_k, t_k)$ is true if and only if $u_k$'s memory is updated at time $t_k$. For example, in Figure 1, the memory of node 6 is updated by events $e_3$ and $e_6$, thus, the $\mathcal{H}(6, t_6) = \{e_3 = (5, 6, t_3), e_6 = (4, 6, t_6)\}$. Based on the $\mathcal{H}(u_k, t)$, we construct memory backtracking trees and assign event contributions. The recursive procedure is given in Algorithm 2, where the maximum depth $L$ controls how far the backtracking proceeds: larger $L$ allows deeper tracing into historical events.

---

**Algorithm 2** Construction of the memory backtracking tree and recursive relevance propagation

---

1: **Input**: target node $u_k$, node memory contributions $\mathbf{M}^t_{u_k \to u_k}$, time $t$, $\mathbf{C}^t_{u_k}$, and the max depth $L$
2: Initialize tree $\mathcal{T}_{\text{memory}} \leftarrow \{(u_k, t)\}$ with root $(u_k, t)$
3: Initialize chained maps $\mathbf{J}_{\text{src}}, \mathbf{J}_{\text{dst}}, \mathbf{J}_{\text{evt}} \leftarrow \mathbf{I}$
4: **procedure** DFS($u, \tau, T_l, \mathbf{J}_{\text{src}}, \mathbf{J}_{\text{dst}}, \mathbf{J}_{\text{evt}}$)
5:     **if** $T_l \geq L$ **then**                                              ▷ reached depth limit
6:         **return**
7:     **end if**
8:     Let $\mathcal{H}(u, \tau) = \{e_1, \ldots, e_m\}$ sorted by $t_1 < \cdots < t_m \leq \tau$
9:     **if** $m = 0$ **then**                                ▷ leaf: nothing earlier than $\tau$
10:         **return**
11:     **end if**
12:     **for** $j = m$ **down to** 1 **do**                              ▷ latest → earliest
13:         $e_j = (v_j, u, t_j)$
14:         Insert $(v_j, t_j)$, $(u, t_j)$ and $(e_j, t_j)$ as children of $(u, \tau)$ in $\mathcal{T}_{\text{memory}}$
15:         Compute $R\left(\bar{\mathbf{m}}^{t_j}_u \to \mathbf{s}^{t_j}_u\right)$ according to the Eq. (10)
16:         Obtain $\mathbf{R}_1 = R(\mathbf{s}^{t_j-}_u \to \mathbf{s}^{t_j}_u)$, $R(\mathbf{s}^{t_j-}_v \to \mathbf{s}^{t_j}_u)$, $R(\mathbf{x}_e \to \mathbf{s}^{t_j}_u)$ via Eq. (12)
17:         Compute $\mathbf{R}_2 = R\left(\mathbf{s}^{t_j-}_u \to \mathbf{s}^{t_j}_u\right)$ according to the Eq. (11)
18:         $\mathbf{C}^t_{u_k}[e_j] \leftarrow \mathbf{C}^t_{u_k}[e_j] + \mathbf{1}^\top \mathbf{J}_{\text{evt}} \mathbf{M}^t_{u_k}$
19:         $\mathbf{J}'_{\text{src}} \leftarrow R(\mathbf{s}^{t_j-}_v \to \mathbf{s}^{t_j}_u) \mathbf{J}_{\text{src}}$,   $\mathbf{J}'_{\text{dst}} \leftarrow (\mathbf{R}_1 + \mathbf{R}_2) \mathbf{J}_{\text{dst}}$,   $\mathbf{J}'_{\text{evt}} \leftarrow R(\mathbf{x}_e \to \mathbf{s}^{t_j}_u) \mathbf{J}_{\text{evt}}$
20:         DFS($v_j, t_j, \mathbf{J}'_{\text{src}}, \mathbf{J}'_{\text{dst}}, \mathbf{J}'_{\text{evt}}$)
21:         DFS($u, t_j, \mathbf{J}'_{\text{src}}, \mathbf{J}'_{\text{dst}}, \mathbf{J}'_{\text{evt}}$)
22:     **end for**
23: **end procedure**
24: DFS($u_k, t, 0, \mathbf{J}_{\text{src}}, \mathbf{J}_{\text{dst}}, \mathbf{J}_{\text{evt}}$)
25: **Output:** $\mathcal{T}_{\text{memory}}(u_k, t)$, contributions $\mathbf{C}^t_{u_k}$

---

## 4.4 SELECTING IMPORTANT EVENTS

We derive the KL divergence and formulate an optimization problem to select important events that faithfully explain the model's behavior. For the link prediction task, the predicted probability is $\hat{y}_{v_k, u_k} = \sigma(f_{\text{mlp}}([\mathbf{z}^t_{v_k} || \mathbf{z}^t_{u_k}]))$. Using the Algorithm 1 and Algorithm 2, we can compute the contributions $\mathbf{C}^t_{u_k}$ and $\mathbf{C}^t_{v_k}$. Then, we update $\mathbf{C}^t_{u_k}$ and $\mathbf{C}^t_{v_k}$ according to Eq (6). Due to the conservation property of LRP, we have $\mathbf{1}^\top \mathbf{C}^t_{u_k} + \mathbf{1}^\top \mathbf{C}^t_{v_k} = f_{\text{mlp}}([\mathbf{z}^t_{v_k} || \mathbf{z}^t_{u_k}]) = y_{v_k, u_k}$. We merge the two matrices. If an event appears in both matrices, its contribution is added together; otherwise, it is retained as is, resulting in the final contribution matrix $\mathbf{C}^t$.

For the link prediction, let $p_1 = \hat{y}_{v_k, u_k}\big(\mathcal{E}_1(t)\big)$ and $p_2 = \hat{y}_{v_k, u_k}\big(\mathcal{E}_2(t)\big)$ denote the probability that nodes $v_k$ and $u_k$ have an interaction on events $\mathcal{E}_1(t)$ and $\mathcal{E}_2(t)$, respectively. Let $z_1 = y_{v_k, u_k}\big(\mathcal{E}_1(t)\big)$ and $z_2 = y_{v_k, u_k}\big(\mathcal{E}_2(t)\big)$ denote the logits. The KL divergence between $p_1$ and $p_2$ is defined as:

$$\text{KL}(p_2 \parallel p_1) = p_2 \log\left(\frac{p_2}{p_1}\right) + (1 - p_2) \log\left(\frac{1 - p_2}{1 - p_1}\right) = \sigma(z_2)\left(z_2 - \text{sp}(z_2) - z_1 - \text{sp}(z_1)\right)$$

$$+ (1 - \sigma(z_2))\left(-\text{sp}(z_2) + \text{sp}(z_1)\right) = \sigma(z_2)(z_2 - z_1) - \text{sp}(z_2) + \text{sp}(z_1), \tag{13}$$

where $\text{sp}(z) = \frac{1}{1 + e^{-z}}$, $\sigma$ is the sigmoid function.

Let $\mathcal{E}^* \subset \mathcal{E}(t)$ be the selected important events, and we aim to minimize the KL divergence between $\hat{y}_{v_k, u_k}\big(\mathcal{E}(t)\big)$ and $\hat{y}_{v_k, u_k}\big(\mathcal{E}^*\big)$. Let $d_c$ denote the number of events in $\mathbf{C}^t$, and $\mathbf{d} \in \{0, 1\}^{d_c}$ be the selection vector, where the element $d_l$ in $\mathbf{d}$ indicates whether the $l$-th event $e_l$ is selected. We define the following optimization problem:

$$\mathbf{d}^* = \underset{\mathbf{d} \in \{0,1\}^{d_c}, \|\mathbf{d}\|_1 = n}{\arg\min} -\hat{y}_{v_k, u_k}\big(\mathcal{E}(t)\big) \sum_{l=1}^{d_c} d_l \mathbf{C}^t_{e_l} + \mathbf{sp}(\sum_{l=1}^{d_c} d_l \mathbf{C}^t_{e_l}) \tag{14}$$

By solving Eq. (14), we can obtain the most important events for the link prediction task. The Algorithm 3 shows the overall process of selecting important events for link prediction task. The selection for node property prediction task is shown in Appendix A.4. The overall process of selecting important events for node property prediction task is shown in Algorithm 4.

---

**Algorithm 3** The overall framework for link prediction

---

1: **Input**: target interaction $(v_k, u_k)$, time $t$, and the max depth of memory backtracking tree $L$
2: Compute $\mathcal{T}_{\text{topology}}(u_k)$, $\mathcal{T}_{\text{topology}}(v_k)$, $\mathbf{C}_{u_k}^t$, $\mathbf{C}_{v_k}^t$, $\{\mathbf{M}_{p \to u_k}^t | p$ is a leaf node of $\mathcal{T}_{\text{topology}}(u_k)\}$, and
$\{\mathbf{M}_{p \to v_k}^t | p$ is a leaf node of $\mathcal{T}_{\text{topology}}(v_k)\}$ using Algorithm 1. $\qquad \triangleright$ topology attribution
3: **for** each $\mathbf{M}_{p \to u_k}^t \in \{\mathbf{M}_{p \to u_k}^t | p$ is a leaf node of $\mathcal{T}_{\text{topology}}(u_k)\}$ **do**
4: $\qquad$ Update $\mathbf{C}_{u_k}^t$ using Algorithm 2 $\qquad\qquad\qquad\qquad\qquad \triangleright$ Memory attribution
5: **end for**
6: **for** each $\mathbf{M}_{p \to v_k}^t \in \{\mathbf{M}_{p \to v_k}^t | p$ is a leaf node of $\mathcal{T}_{\text{topology}}(v_k)\}$ **do**
7: $\qquad$ Update $\mathbf{C}_{v_k}^t$ using Algorithm 2 $\qquad\qquad\qquad\qquad\qquad \triangleright$ Memory attribution
8: **end for**
9: Update $\mathbf{C}_{u_k}^t$ and $\mathbf{C}_{v_k}^t$ using Eq. (6), and merge $\mathbf{C}_{u_k}^t$ and $\mathbf{C}_{v_k}^t$ to obtain $\mathbf{C}^t$
10: Select important events $\mathcal{E}^*$ using Eq. (14)
11: **Output:** The important events $\mathcal{E}^*$

---

## 5 EXPERIMENTS

**Datasets**. We evaluate the effectiveness of our method, MemExplainer, on several real-world temporal graph datasets: Wikipedia, Reddit, Enron and UCI (Hamilton et al., 2017; Jure, 2014; Poursafaei et al., 2022) for the link prediction task, and tgbn-trade, tgbn-genre, and tgbn-reddit (Huang et al., 2023) for node property prediction. Wikipedia and Reddit are bipartite networks with rich interaction attributes, while Enron and UCI are social networks without interaction attributes. tgbn-trade represents an international agriculture trade network, tgbn-genre is a bipartite, weighted interaction network between users and music genres, tgbn-reddit captures user-subreddit interactions. Detailed dataset statistics are provided in Appendix A.6.

**Baselines**. We compare the performance of MemExplainer with several baselines: GNNExplainer (Ying et al., 2019), PGExplainer (Luo et al., 2020), TGNNExplainer (Xia et al., 2022), and TempME (Chen & Ying, 2023). The GNNExplainer and PGExplainer are the explanation methods for static graph models. We adapt it for TGNs following the same setting in prior work (Xia et al., 2022). The TGNNExplainer and TempME are current state-of-the-art model specifically designed for TGNs. TempME introduces a graph generation approach to capture temporal motifs for TGN predictions, while TGNNExplainer combines a navigator with Monte Carlo Tree Search, guiding the sampling process to construct explanation subgraphs. We also include several ablation variants as baselines. **w/o memory** uses only neighboring-event contributions (no memory backtracking tree); **w/o topology** uses only historical-event contributions (no topology tree); **w/o selection** skips objective-based selection and directly returns the top-$k$ events by accumulated contribution.

**Evaluation metrics**. We adopt the fidelity and the sparsity to evaluate the performance. Let $\mathcal{E}(t)$ represent the original temporal events. Let $\mathcal{E}^*$ denote the selected important events. The KL-based fidelity metrics are $\text{Fidelity}_{\text{KL}} = \text{KL}(\hat{y}_{v_k, u_k}(\mathcal{E}(t)) \| \hat{y}_{v_k, u_k}(\mathcal{E}^*))$ (link prediction), and $\text{Fidelity}_{\text{KL}} = \text{KL}(\hat{y}_{u_k}(\mathcal{E}(t)) \| \hat{y}_{u_k}(\mathcal{E}^*))$ (node property prediction) as defined in prior work (Liu et al., 2024). We also use the probability-based fidelity metric: $\text{Fidelity}_{\text{prob}}$: $\text{Fidelity}_{\text{prob}} = |\hat{y}_{v_k, u_k}(\mathcal{E}(t)) - \hat{y}_{v_k, u_k}(\mathcal{E}^*)|$ (link prediction) and $\text{Fidelity}_{\text{prob}} = |\hat{y}_{u_k}(\mathcal{E}(t)) - \hat{y}_{u_k}(\mathcal{E}^*)|$ (node property prediction), as defined in (Yuan et al., 2022). The lower, the better. Lower values indicate better performance. Sparsity $= \frac{|\mathcal{E}^*|}{|\mathcal{E}(t)|}$, where $|\mathcal{E}^*|$ and $|\mathcal{E}(t)|$ denote the number of events in $\mathcal{E}^*$ and $\mathcal{E}(t)$, respectively.

**Experimental setup**. We train TGNs with a 70%, 15%, and 15% splitting scheme of datasets based on timestamps. We set the maximum depth of the memory backtracking tree to 5. Given the target events or nodes, we apply Algorithm 3 or Algorithm 4 to identify the important events $\mathcal{E}^*$.

**Performance**. We report the $\text{Fidelity}_{\text{KL}}$ and the $\text{Fidelity}_{\text{prob}}$ results in Figure 2 and Figure 3, respectively. Our method, MemExplainer, significantly outperforms baseline explainers across all

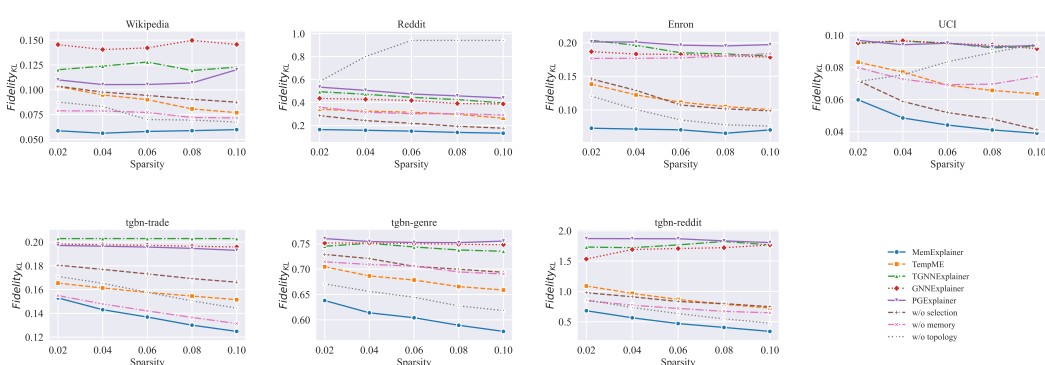

Figure 2: The performance of Fidelity$_{KL}$. Each figure corresponds to a different dataset. First and second rows represent link prediction and node property prediction, respectively. Lower value indicates better performance.

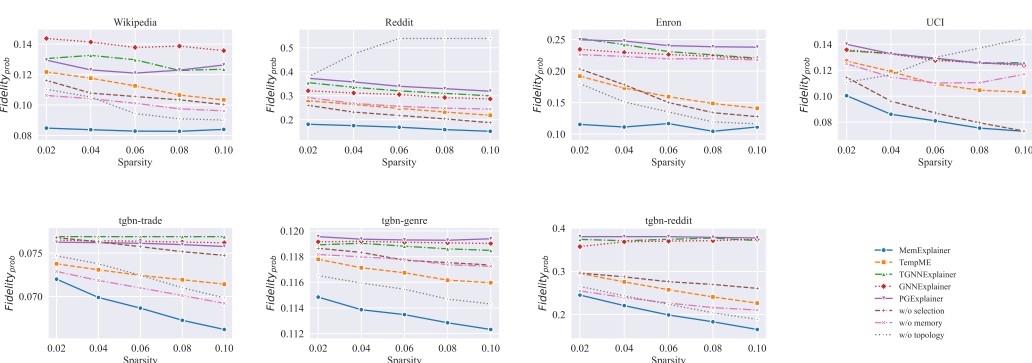

Figure 3: The performance of Fidelity$_{prob}$. Each figure corresponds to a different dataset. First and second rows represent link prediction and node property prediction, respectively. Lower value indicates better performance.

datasets for both metrics, demonstrating its ability to maintain high fidelity in explanations even with low sparsity. TempME can only capture temporal motifs and does not trace node memory vectors. TGNNExplainer relies on sampling; when many events are present, it only considers the sampled candidates, and if these have little influence, the resulting explanations have low fidelity. The performances of GNNExplainer and PGExplainer are poorer, as these methods are designed for static graphs and are not suitable for TGNs. Some of our variant baselines outperform TempME on Wikipedia, tgbn-reddit datasets. We also perform a sensitivity analysis of the depth of the memory backtracking tree, detailed in Appendix A.7. As shown in Figures 4 and 5, Fidelity$_{KL}$ and Fidelity$_{prob}$ first decrease and then increase with depth. Deeper backtracking extends the trace duration but expands the search space, complicating event selection and reducing fidelity.

## 6 CONCLUSIONS

We study the problem of explaining predictions in TGNs, addressing the limitations of prior work that primarily focus on graph topology while overlooking the temporal evolution of node memories. We propose a topology attribution tree to attribute TGNs logits to recent events and node memories. Then we propose a memory backtracking tree to propagate memory contributions to historical events that responsible for memory updates. Due to the properties of LRP, our method ensure that the total contribution of events equals the logits of TGNs. We also derive the KL divergence and design an objective function to select important events as explanations. Experiments on node property prediction and link prediction tasks show that our method outperforms baselines across all datasets, highlighting the advantage of tracing memory vectors for better explanations.

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

# A  APPENDIX

## A.1  THE USE OF LARGE LANGUAGE MODELS (LLMs)

The Large Language Models (LLMs) were used solely for the purpose of polishing the manuscript. Specifically, the LLM was employed to assist with tasks such as grammar correction, and spelling error detection. The LLM was used exclusively to enhance the clarity and fluency of the writing.

## A.2  DERIVATION OF LRP APPLIED TO $f_{\text{emb}}$

Applying LRP to the first equation in Eq. (7) yields the following scalar formulations:

$$R\big(\mathbf{h}_{u_k}^{t,l-1}[i] \rightarrow \mathbf{h}_{u_k}^{t,l}[j]\big) = \frac{\mathbf{h}_{u_k}^{t,l-1}[i]\,\mathbf{W}_2^l[i,j]}{\mathbf{h}_{u_k}^{t,l}[j]}\,R\big(\mathbf{h}_{u_k}^{t,l}[j]\big)\,, \tag{15}$$

$$R\Big(\tilde{\mathbf{h}}_{u_k}^{t,l}[i] \rightarrow \mathbf{h}_{u_k}^{t,l}[j]\Big) = \frac{\tilde{\mathbf{h}}_{u_k}^{t,l}[i]\,\mathbf{W}_2^l[d_m+i,\,j]}{\mathbf{h}_{u_k}^{t,l}[j]}\,R\big(\mathbf{h}_{u_k}^{t,l}[j]\big)\,, \tag{16}$$

where $R(\mathbf{h}_{u_k}^{t,l-1}[i] \rightarrow \mathbf{h}_{u_k}^{t,l}[j])$ denotes the relevance contribution of the $i$-th elements of $\mathbf{h}_{u_k}^{t,l-1}$ to the $j$-th elements of $\mathbf{h}_{u_k}^{t,l}$. Similarly, $R(\tilde{\mathbf{h}}_{u_k}^{t,l}[i] \rightarrow \mathbf{h}_{u_k}^{t,l}[j])$ denotes the relevance of the $i$-th elements of $\tilde{\mathbf{h}}_{u_k}^{t,l-1}$ to the $j$-th elements of $\mathbf{h}_{u_k}^{t,l}$, $d_m$ and $d_n$ is the dimension of the $\mathbf{h}_{u_k}^{t,l-1}$ and $\tilde{\mathbf{h}}_{u_k}^{t,l}$, respectively.

We rewrite Eq. (15)-(16) in matrix form. Let $\mathbf{A}^l = \mathbf{W}_2^l[1:d_m,:]$, $\mathbf{B}^l = \mathbf{W}_2^l[d_m+1:\text{end},:]$. Then,

$$R\big(\mathbf{h}_{u_k}^{t,l-1} \rightarrow \mathbf{h}_{u_k}^{t,l}\big) = \text{diag}\big(\mathbf{h}_{u_k}^{t,l-1}\big)\,\mathbf{A}^l\,\text{diag}(R\big(\mathbf{h}_{u_k}^{t,l}\big) \oslash \mathbf{h}_{u_k}^{t,l})$$

$$R\Big(\tilde{\mathbf{h}}_{u_k}^{t,l} \rightarrow \mathbf{h}_{u_k}^{t,l}\Big) = \text{diag}\big(\tilde{\mathbf{h}}_{u_k}^{t,l}\big)\,\mathbf{B}^l\,\text{diag}\big(R\big(\mathbf{h}_{u_k}^{t,l}\big) \oslash \mathbf{h}_{u_k}^{t,l}\big)\,.$$

Where $\text{diag}(\cdot)$ denotes the diagonalization operator that maps a vector to a diagonal matrix, $\oslash$ represents element-wise division. Accordingly, the resulting matrices $R\big(\mathbf{h}_{u_k}^{t,l-1} \rightarrow \mathbf{h}_{u_k}^{t,l}\big) \in \mathbb{R}^{d_m \times d_m}$ and $R\Big(\tilde{\mathbf{h}}_{u_k}^{t,l} \rightarrow \mathbf{h}_{u_k}^{t,l}\Big) \in \mathbb{R}^{d_n \times d_m}$ quantify the relevance propagated from each input dimension to each output dimension.

Let $\hat{\mathbf{h}}_{u_k}^{t,l} = \sum_{e_k \in \mathcal{N}_{u_k}^n([0,t))}\Big(\mathbf{h}_{v_k}^{t,l-1} \,\|\, \mathbf{x}_{e_k} \,\|\, \phi(t-t_k)\Big)\mathbf{W}_1^l$, we apply LRP on the second equation in Eq. (7), we can obtain the following equation:

$$R\Big(\mathbf{h}_{v_k}^{t,l-1}[i] \rightarrow \tilde{\mathbf{h}}_{u_k}^{t,l}[j]\Big) = \frac{\mathbf{h}_{v_k}^{t,l-1}[i]\,\mathbf{W}_1^l[i,\,j]}{\hat{\mathbf{h}}_{u_k}^{t,l}[j]}\,R\Big(\tilde{\mathbf{h}}_{u_k}^{t,l}[j]\Big)\,,$$

$$R\Big(\mathbf{x}_{e_k}[i] \rightarrow \tilde{\mathbf{h}}_{u_k}^{t,l}[j]\Big) = \frac{\mathbf{x}_{e_k}[i]\,\mathbf{W}_1^l[d_m+i,\,j]}{\hat{\mathbf{h}}_{u_k}^{t,l}[j]}\,R\Big(\tilde{\mathbf{h}}_{u_k}^{t,l}[j]\Big)\,,$$

$$R\Big(\phi(t-t_k)[i] \rightarrow \tilde{\mathbf{h}}_{u_k}^{t,l}[j]\Big) = \frac{\phi(t-t_k)[i]\,\mathbf{W}_1^l[d_m+d_e+i,\,j]}{\hat{\mathbf{h}}_{u_k}^{t,l}[j]}\,R\Big(\tilde{\mathbf{h}}_{u_k}^{t,l}[j]\Big)\,,$$

where $d_e$ is the dimension of $\mathbf{x}_{e_k}$.

We rewrite the above equation in matrix form. Let $\mathbf{C}^l = \mathbf{W}_1^l[1:d_m,:]$, $\mathbf{D}^l = \mathbf{W}_1^l[d_m+1:d_m+d_e,:]$, $\mathbf{E}^l = \mathbf{W}_2^l[d_m+d_e+1:\text{end},:]$, then

$$R\Big(\mathbf{h}_{v_k}^{t,l-1} \rightarrow \tilde{\mathbf{h}}_{u_k}^{t,l}\Big) = \text{diag}\big(\mathbf{h}_{v_k}^{t,l-1}\big)\,\mathbf{C}^l\,\text{diag}\Big(R\,\Big(\tilde{\mathbf{h}}_{u_k}^{t,l}[j]\Big) \oslash \hat{\mathbf{h}}_{u_k}^{t,l}\Big)$$

$$R\Big(\mathbf{x}_{e_k} \rightarrow \tilde{\mathbf{h}}_{u_k}^{t,l}\Big) = \text{diag}\big(\mathbf{x}_{e_k}\big)\,\mathbf{D}^l\,\text{diag}\Big(R\,\Big(\tilde{\mathbf{h}}_{u_k}^{t,l}[j]\Big) \oslash \hat{\mathbf{h}}_{u_k}^{t,l}\Big)$$

$$R\Big(\phi(t-t_k) \rightarrow \tilde{\mathbf{h}}_{u_k}^{t,l}\Big) = \text{diag}\big(\phi(t-t_k)\big)\,\mathbf{E}^l\,\text{diag}\Big(R\,\Big(\tilde{\mathbf{h}}_{u_k}^{t,l}[j]\Big) \oslash \hat{\mathbf{h}}_{u_k}^{t,l}\Big)\,.$$

Thus,

$$R\big(\mathbf{h}_{v_k}^{t,l-1} \to \mathbf{h}_{u_k}^{t,l}\big) = \sum_p \frac{\mathbf{h}_{v_k}^{t,l-1}[i]\,\mathbf{W}_1^l[i,\,p]}{\hat{\mathbf{h}}_{u_k}^{t,l}[p]}\,\frac{\tilde{\mathbf{h}}_{u_k}^{t,l}[p]\,\mathbf{W}_2^l[d_m+p,\,j]}{\mathbf{h}_{u_k}^{t,l}[j]}\,R\big(\mathbf{h}_{u_k}^{t,l}[j]\big),$$

$$R\big(\mathbf{x}_{e_k}[i] \to \mathbf{h}_{u_k}^{t,l}[j]\big) = \sum_p \frac{\mathbf{x}_{e_k}[i]\,\mathbf{W}_1^l[d_m+i,\,p]}{\hat{\mathbf{h}}_{u_k}^{t,l}[p]}\,\frac{\tilde{\mathbf{h}}_{u_k}^{t,l}[p]\,\mathbf{W}_2^l[d_m+p,\,j]}{\mathbf{h}_{u_k}^{t,l}[j]}\,R\big(\mathbf{h}_{u_k}^{t,l}[j]\big),$$

$$R\big(\phi(t-t_k)[i] \to \mathbf{h}_{u_k}^{t,l}[j]\big) = \sum_p \frac{\tilde{\mathbf{h}}_{u_k}^{t,l}[p]\,\mathbf{W}_2^l[d_m+p,\,j]}{\mathbf{h}_{u_k}^{t,l}[j]}\,\frac{\phi(t-t_k)[i]\,\mathbf{W}_1^l[d_m+d_e+i,\,p]}{\hat{\mathbf{h}}_{u_k}^{t,l}[p]}\,R\big(\mathbf{h}_{u_k}^{t,l}[j]\big),$$

We can also rewrite the above equation in the matrix form:

$$R\big(\mathbf{h}_{v_k}^{t,l-1} \to \mathbf{h}_{u_k}^{t,l}\big) = \mathrm{diag}\big(\mathbf{h}_{v_k}^{t,l-1}\big)\,\mathbf{C}^l\,\mathrm{diag}\!\left(\frac{\tilde{\mathbf{h}}_{u_k}^{t,l}}{\hat{\mathbf{h}}_{u_k}^{t,l}}\right)\,\mathbf{B}^l\,\mathrm{diag}\!\left(\frac{R(\mathbf{h}_{u_k}^{t,l})}{\mathbf{h}_{u_k}^{t,l}}\right),$$

$$R\big(\mathbf{x}_{e_k} \to \mathbf{h}_{u_k}^{t,l}\big) = \mathrm{diag}\big(\mathbf{x}_{e_k}\big)\,\mathbf{D}^l\,\mathrm{diag}\!\left(\frac{\tilde{\mathbf{h}}_{u_k}^{t,l}}{\hat{\mathbf{h}}_{u_k}^{t,l}}\right)\,\mathbf{B}^l\,\mathrm{diag}\!\left(\frac{R(\mathbf{h}_{u_k}^{t,l})}{\mathbf{h}_{u_k}^{t,l}}\right),$$

$$R\big(\phi(t-t_k) \to \mathbf{h}_{u_k}^{t,l}\big) = \mathrm{diag}\big(\phi(t-t_k)\big)\,\mathbf{E}^l\,\mathrm{diag}\!\left(\frac{\tilde{\mathbf{h}}_{u_k}^{t,l}}{\hat{\mathbf{h}}_{u_k}^{t,l}}\right)\,\mathbf{B}^l\,\mathrm{diag}\!\left(\frac{R(\mathbf{h}_{u_k}^{t,l})}{\mathbf{h}_{u_k}^{t,l}}\right).$$

Finally, if the temporal graph sum function has $L$ layers, and $R(\mathbf{h}_{u_k}^{t,L}) = \mathbf{h}_{u_k}^{t,L}$

$$R\big(\mathbf{h}_{u_k}^{t,0} \to \mathbf{h}_{u_k}^{t,L}\big) = \mathrm{diag}\big(\mathbf{h}_{u_k}^{t,0}\big)\,\left(\prod_{l=1}^{L}\mathbf{A}^l\right).$$

Given the path $(p_l, p_{l+1}, \ldots, p_L)$, $p_L = u_k$, this path corresponds to $L - l$ events, which are $e_l, e_{l+1}, \cdots, e_{L-1}$, and $e_l = (p_l, p_{l+1}, t_l)$. We define an operator $\mathbf{T}^l(p_l \to p_{l+1}) = \mathbf{C}^{l+1}\,\mathrm{diag}\!\left(\frac{\tilde{\mathbf{h}}_{p_{l+1}}^{t,l+1}}{\hat{\mathbf{h}}_{p_{l+1}}^{t,l+1}}\right)\,\mathbf{B}^{l+1}$, where $p_l \in \mathcal{N}_{p_{l+1}}^n([0,t])$, then the relevance of node $p_l$ to $u_k$ and $e_l$ to $u_k$ is,

$$R\big(\mathbf{h}_{p_l}^{t,l} \to \mathbf{h}_{u_k}^{t,L}\big) = \mathrm{diag}\big(\mathbf{h}_{p_l}^{t,l}\big)\,\left(\prod_{r=l}^{L-1}\mathbf{T}^r(p_r \to p_{r+1})\right)$$

$$R\big(\mathbf{x}_{e_l} \to \mathbf{h}_{u_k}^{t,L}\big) = \mathrm{diag}(\mathbf{x}_{e_l})\,\mathbf{D}^{l+1}\,\mathrm{diag}\!\left(\frac{\tilde{\mathbf{h}}_{p_{l+1}}^{t,l+1}}{\hat{\mathbf{h}}_{p_{l+1}}^{t,l+1}}\right)\,\mathbf{B}^{l+1}\,\left(\prod_{r=l+1}^{L-1}\mathbf{T}^r(p_r \to p_{r+1})\right)$$

$$R\big(\phi(t-t_l) \to \mathbf{h}_{u_k}^{t,l}\big) = \mathrm{diag}(\phi(t-t_l))\,\mathbf{E}^{l+1}\,\mathrm{diag}\!\left(\frac{\tilde{\mathbf{h}}_{p_{l+1}}^{t,l+1}}{\hat{\mathbf{h}}_{p_{l+1}}^{t,l+1}}\right)\,\mathbf{B}^{l+1}\,\left(\prod_{r=l+1}^{L-1}\mathbf{T}^r(p_r \to p_{r+1})\right).$$

### A.3 Derivation of LRP Applied to GRU function

If the $f_{\text{update}}$ is the GRU function, the Eq. (3) can be rewritten as

$$\text{Reset gate:}\quad \mathbf{r}_{u_k}^t = \sigma\big(\bar{\mathbf{m}}_{u_k}^t\mathbf{W}_r + \mathbf{s}_{u_k}^{t-}\mathbf{U}_r^t + \mathbf{b}_r\big),$$

$$\text{Update gate:}\quad \mathbf{g}_{u_k}^t = \sigma\big(\bar{\mathbf{m}}_{u_k}^t\mathbf{W}_g + \mathbf{s}_{u_k}^{t-}\mathbf{U}_g + \mathbf{b}_g\big),$$

$$\text{Candidate state:}\quad \tilde{\mathbf{s}}_{u_k}^t = \tanh\big(\bar{\mathbf{m}}_{u_k}^t\mathbf{W}_h + \mathbf{r}_{u_k}^t \odot (\mathbf{s}_{u_k}^{t-}\mathbf{U}_h) + \mathbf{b}_h\big),$$

$$\text{Updated state:}\quad \mathbf{s}_{u_k}^t = (1 - \mathbf{g}_{u_k}^t) \odot \mathbf{s}_{u_k}^{t-} + \mathbf{g}_{u_k}^t \odot \tilde{\mathbf{s}}_{u_k}^t.$$

Where, $\mathbf{W}_r, \mathbf{W}_g, \mathbf{W}_h, \mathbf{U}_r, \mathbf{U}_g, \mathbf{U}_h, \mathbf{b}_r, \mathbf{b}_g, \mathbf{b}_h$ are the parameters in the GRU.

In GRU computation, multiplicative interactions occur when a gate neuron modulates a signal neuron, e.g., $\mathbf{g}_{u_k}^t[i] \cdot \mathbf{s}_{u_k}^{t-}[i]$. Unlike linear mappings, such interactions pose challenges for relevance redistribution. A widely adopted strategy is the signal-take-all rule, which assigns all relevance to

the signal neuron and none to the gate. This reflects the view that the gate controls the flow of information, but is not information itself (Wu et al., 2022).

Thus,

$$R\big(\mathbf{s}_{u_k}^{t-} \to \mathbf{s}_{u_k}^t\big) = \mathrm{diag}\bigg(\frac{(1 - \mathbf{g}_{u_k}^t) \odot \mathbf{s}_{u_k}^{t-}}{\mathbf{s}_{u_k}^t}\bigg), \tag{17}$$

$$R\big(\tilde{\mathbf{s}}_{u_k}^t \to \mathbf{s}_{u_k}^t\big) = \mathrm{diag}\bigg(\frac{\mathbf{g}_{u_k}^t \odot \tilde{\mathbf{s}}_{u_k}^t}{\mathbf{s}_{u_k}^t}\bigg). \tag{18}$$

Then for the $\tilde{\mathbf{s}}_{u_k}^t$,

$$R\big(\bar{\mathbf{m}}_{u_k}^t \to \tilde{\mathbf{s}}_{u_k}^t\big) = \mathrm{diag}\big(\bar{\mathbf{m}}_{u_k}^t\big) \, \mathbf{W}_h \, \mathrm{diag}\bigg(\frac{R\big(\tilde{\mathbf{s}}_{u_k}^t \to \mathbf{s}_{u_k}^t\big)}{\bar{\mathbf{m}}_{u_k}^t \mathbf{W}_h + \mathbf{r}_{u_k}^t \odot (\mathbf{s}_{u_k}^{t-} \mathbf{U}_h)}\bigg), \tag{19}$$

$$R\big(\mathbf{s}_{u_k}^{t-} \to \tilde{\mathbf{s}}_{u_k}^t\big) = \mathrm{diag}\big(\mathbf{s}_{u_k}^{t-}\big) \, \mathbf{U}_h \, \mathrm{diag}\bigg(\frac{\mathbf{r}_{u_k}^t \odot R\big(\tilde{\mathbf{s}}_{u_k}^t \to \mathbf{s}_{u_k}^t\big)}{\bar{\mathbf{m}}_{u_k}^t \mathbf{W}_h + \mathbf{r}_{u_k}^t \odot (\mathbf{s}_{u_k}^{t-} \mathbf{U}_h)}\bigg). \tag{20}$$

Finally,

$$R\big(\bar{\mathbf{m}}_{u_k}^t \to \mathbf{s}_{u_k}^t\big) = \mathrm{diag}\big(\bar{\mathbf{m}}_{u_k}^t\big) \, \mathbf{W}_h \, \mathrm{diag}\Bigg(\frac{\mathbf{g}_{u_k}^t \odot \mathrm{diag}\big(\frac{\mathbf{g}_{u_k}^t \odot \tilde{\mathbf{s}}_{u_k}^t}{\mathbf{s}_{u_k}^t}\big)}{\bar{\mathbf{m}}_{u_k}^t \mathbf{W}_h + \mathbf{r}_{u_k}^t \odot (\mathbf{s}_{u_k}^{t-} \mathbf{U}_h)}\Bigg),$$

$$R\big(\mathbf{s}_{u_k}^{t-} \to \mathbf{s}_{u_k}^t\big) = \mathrm{diag}\big(\mathbf{s}_{u_k}^{t-}\big) \, \mathbf{U}_h \, \mathrm{diag}\Bigg(\frac{\mathbf{r}_{u_k}^t \odot \mathbf{g}_{u_k}^t \odot \mathrm{diag}\big(\frac{\mathbf{g}_{u_k}^t \odot \tilde{\mathbf{s}}_{u_k}^t}{\mathbf{s}_{u_k}^t}\big)}{\bar{\mathbf{m}}_{u_k}^t \mathbf{W}_h + \mathbf{r}_{u_k}^t \odot (\mathbf{s}_{u_k}^{t-} \mathbf{U}_h)}\Bigg)$$

$$+ \, \mathrm{diag}\bigg(\frac{(1 - \mathbf{g}_{u_k}^t) \odot \mathbf{s}_{u_k}^{t-}}{\mathbf{s}_{u_k}^t}\bigg).$$

Where $\mathrm{diag}(\cdot)$ denotes the diagonalization operator that maps a vector to a diagonal matrix, $-$ represents element-wise division. $R\big(\mathbf{s}_{u_k}^{t-} \to \mathbf{s}_{u_k}^t\big) \in \mathbb{R}^{d_m \times d_m}$ and $R\big(\bar{\mathbf{m}}_{u_k}^t \to \mathbf{s}_{u_k}^t\big) \in \mathbb{R}^{d_m' \times d_m}$, where $d_m'$ is the dimension of $\bar{\mathbf{m}}_{u_k}^t$.

Let $\mathbf{F}^t = \mathbf{g}_{u_k}^t \odot \mathrm{diag}\big(\frac{\mathbf{g}_{u_k}^t \odot \tilde{\mathbf{s}}_{u_k}^t}{\mathbf{s}_{u_k}^t}\big)$, $\mathbf{H}^t = \bar{\mathbf{m}}_{u_k}^t \mathbf{W}_h + \mathbf{r}_{u_k}^t \odot (\mathbf{s}_{u_k}^{t-} \mathbf{U}_h)$

$$R\big(\bar{\mathbf{m}}_{u_k}^t \to \mathbf{s}_{u_k}^t\big) = \mathrm{diag}\big(\bar{\mathbf{m}}_{u_k}^t\big) \, \mathbf{W}_h \, \mathrm{diag}\bigg(\frac{\mathbf{F}^t}{\mathbf{H}^t}\bigg), \tag{21}$$

$$R\big(\mathbf{s}_{u_k}^{t-} \to \mathbf{s}_{u_k}^t\big) = \mathrm{diag}\big(\mathbf{s}_{u_k}^{t-}\big) \, \mathbf{U}_h \, \mathrm{diag}\bigg(\frac{\mathbf{r}_{u_k}^t \odot \mathbf{F}^t}{\mathbf{H}^t}\bigg) + \mathrm{diag}\bigg(\frac{(1 - \mathbf{g}_{u_k}^t) \odot \mathbf{s}_{u_k}^{t-}}{\mathbf{s}_{u_k}^t}\bigg). \tag{22}$$

### A.4 THE SELECTION FOR NODE PROPERTY PREDICTION TASK.

Similarly, for the node property prediction, the predicted probability is $\hat{\mathbf{y}} = \hat{\mathbf{y}}_{u_k} = \mathrm{softmax}(f_{\mathrm{mlp}}(\mathbf{z}_{u_k}^t))$. Let $\mathbf{z}$ denote the logits, and the contribution matrix of events is $\mathbf{C}_{u_k}^t$. The KL divergence is

$$\mathrm{KL}(\hat{\mathbf{y}}(\mathcal{E}_1(t)) \| \hat{\mathbf{y}}(\mathcal{E}_2(t))) = \sum_{k=1}^{c} \hat{y}_k(\mathcal{E}_2(t)) \log[\frac{\hat{y}_k(\mathcal{E}_2(t))}{\hat{y}_k(\mathcal{E}_1(t))}]$$

$$= \sum_{k=1}^{c} \hat{y}_k(\mathcal{E}_2(t))[z_k(\mathcal{E}_2(t)) - z_k(\mathcal{E}_1(t))] - \log(Z) = \sum_{k=1}^{c} \hat{y}_k(\mathcal{E}_2(t)) \Delta z_k - \log(Z), \tag{23}$$

where $Z(\mathcal{E}_\tau(t)) = \sum_{k=1}^{c} \exp(z_k(\mathcal{E}_\tau(t)))$ for $\tau = 1, 2$, and $Z = \frac{Z(\mathcal{E}_2(t))}{Z(\mathcal{E}_1(t))}$. The objective function for node property prediction is:

$$\mathbf{d}^* = \underset{\mathbf{d} \in \{0,1\}^{d_c}, \|\mathbf{d}\|_1 = n}{\arg\min} \sum_{k=1}^{c} \left( -\hat{y}_k(\mathcal{E}(t)) \sum_{l=1}^{|d_c|} d_l \mathbf{C}_{e_l}^t \right) + \log \sum_{k'=1}^{c} \exp \left( \sum_{l=1}^{|d_c|} d_l \mathbf{C}_{e_l}^t \right) \quad (24)$$

Solving this equation yields the most important events for the node property prediction task. The Algorithm shows the overall process of selecting important layer edges for node property prediction task.

## A.5 THE OVERALL FRAMEWORK FOR NODE PROPERTY PREDICTION TASK

---

**Algorithm 4** The overall framework for node property prediction

---

1: **Input**: target node $u_k$, time $t$, and the max depth of memory backtracking tree $L$
2: Compute $\mathcal{T}_{\text{topology}}(u_k)$, and $\mathcal{T}_{\text{topology}}(v_k)$, $\mathbf{C}_{u_k}^t$ using Algorithm 1.   ▷ topology attribution
3: **for** each $\mathbf{M}_{p \to u_k}^t \in \{\mathbf{M}_{p \to u_k}^t | p \text{ is a leaf node of } \mathcal{T}_{\text{topology}}(u_k)\}$ **do**
4:     Update $\mathbf{C}_{u_k}^t$ using Algorithm 2   ▷ Memory attribution
5: **end for**
6: Update $\mathbf{C}_{u_k}^t$ using Eq. (6), and let $\mathbf{C}_{u_k}^t \to \mathbf{C}^t$
7: Select important events $\mathcal{E}^*$ using Eq. (24)
8: **Output:** The important events $\mathcal{E}^*$

---

## A.6 DATASET

We select several real-world temporal graph datasets for link prediction. These six datasets cover a wide range of real-world applications and domains, including social networks, political networks, communication networks, etc. The brief introduction of the six datasets is listed as follows. Data statistics are given in Table 1

- Wikipedia (Kumar et al., 2019): A one-month interaction network with nodes as editors and pages, and edges as timestamped posting requests.

- Reddit (Kumar et al., 2019): Records one-month activity in subreddits, with nodes as users/posts and edges as timestamped posts. Edge features are 172-dimensional LIWC vectors from edit texts.

- Enron (Shetty & Adibi, 2004): Email interaction network of ENRON employees over three years, with no attributes.

- UCI (Panzarasa et al., 2009): Unattributed social network among UCI students, tracking online forum posts with second-level timestamps.

- tgbn-trade (Huang et al., 2023): International agriculture trade network between UN nations (1986-2016), with edges representing annual trade values.

- tgbn-genre (Huang et al., 2023): Bipartite network between users and music genres, with edge weights denoting genre preferences.

- tgbn-reddit (Huang et al., 2023): Interaction network between users and subreddits (2005-2019), with edges representing posts.

## A.7 SENSITIVITY ANALYSIS OF THE MAXIMUM DEPTH OF THE MEMORY BACKTRACKING TREE

On the Enron, Reddit, UCI, and tgbn-genre datasets, we set the maximum depth of the memory backtracking tree from 2 to 10, selecting the same number of events for each target node or edge across all depths. The performance of FidelityKL and Fidelityprob is shown in Figures 4 and 5, respectively. As shown in these figures, $\text{Fidelity}_{\text{KL}}$ initially decreases and then increases with increasing maximum depth. This trend occurs because deeper backtracking allows for more accurate

Table 1: The details of datasets.

| Datasets | Domains | Nodes | Links |
|----------|---------|-------|-------|
| Wikipedia | Social | 9,227 | 157,474 |
| Reddit | Social | 10,984 | 672,447 |
| Enron | Communication | 184 | 125,235 |
| UCI | Social | 1,899 | 59,835 |
| tgbn-trade | trade | 255 | 468,245 |
| tgbn-genre | interact. | 1,505 | 17,858,395 |
| tgbn-reddit | social | 11,766 | 27,174,118 |

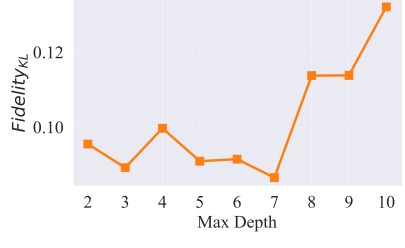
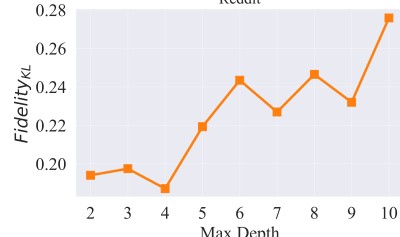
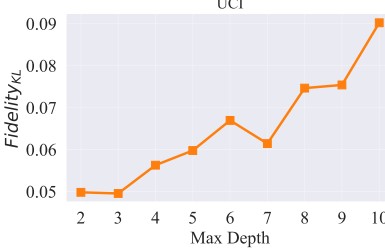
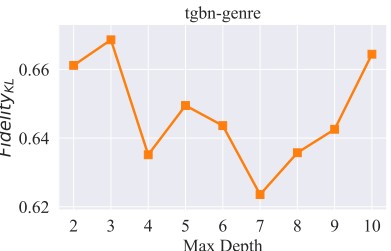

Figure 4: Performance of $\text{Fidelity}_{\text{KL}}$ at different maximum depths of the memory backtracking tree, with each figure representing a different dataset.

tracing of historical events, resulting in more faithful explanations. However, this also increases the number of candidate explanation events, making the optimization problem more complex. The expanded search space complicates the identification of relevant events, leading to the decrease in fidelity. The best performing on different datasets is different, which is due to the different characteristics of the datasets. The optimal maximum depth varies across different datasets, which can be attributed to the unique characteristics of each dataset.

## A.8 CASE STUDY

We evaluate TGN on a pose-based action classification task. For each video, we first run YOLO-Pose Maji et al. (2022) to detect human keypoints and construct a skeleton graph whose nodes are body joints and whose edges follow the human kinematic structure. This produces a sequence of skeleton graphs, one per frame, which we feed into TGN to predict the action label. We use the HMDB51 dataset Kuehne et al. (2011), selecting four classes: sit-up, pull-up, climb, and run, as our train and evaluation subset.

For a fair comparison, all methods operate on exactly the same input sequence of skeleton graphs for each video. We also fix a global edge budget, i.e., the total number of key human-pose edges that each method is allowed to select for that video. In addition to $\text{Fidelity}_{\text{prob}}$ and $\text{Fidelity}_{\text{kl}}$, we introduce

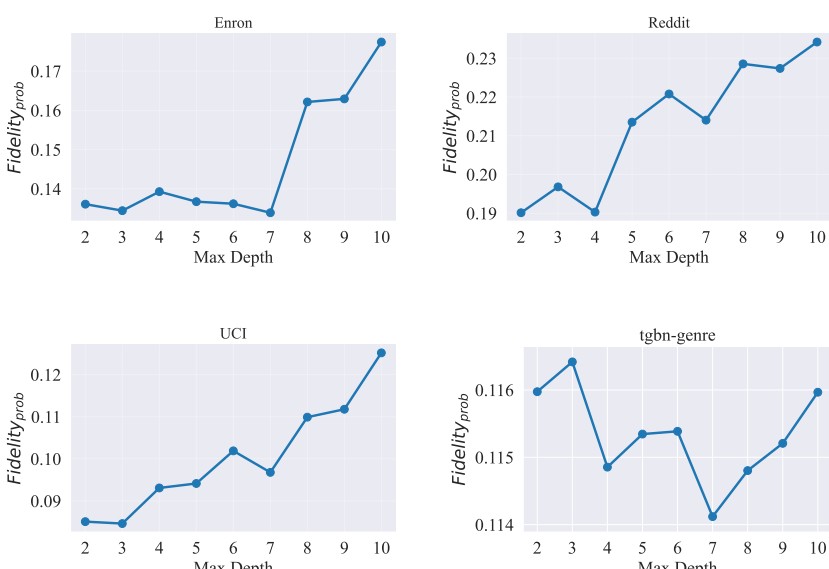

Figure 5: Performance of Fidelity$_{\text{prob}}$ at different maximum depths of the memory backtracking tree, with each figure representing a different dataset.

Fidelity$_{\text{label}}$, which is set to 1 if the label predicted by TGN using only the explained edges matches the true label, and 0 otherwise; we then average this value over all videos. Figure 6 shows the performance of Fidelity$_{\text{prob}}$, Fidelity$_{\text{kl}}$, and Fidelity$_{\text{label}}$. A higher Fidelity$_{\text{label}}$ and lower Fidelity$_{\text{prob}}$ and Fidelity$_{\text{kl}}$ indicate better explanations. As shown in Figure 6, our method consistently outperforms the baselines on the pose-based action classification task.

Tables 2, Table 3, Table 4, and Table 5 show case studies for the pull-up, run, climb, and sit-up classes. Green nodes denote joint (bone) nodes, red edges denote the original connections between joints, yellow edges denote the selected important edges, and each image is one frame from video. Because different methods select different explanation edges, the explanation edges naturally fall on different frames and involve different bone connections. Consequently, these explanation results are not meant to be aligned frame-by-frame across methods.

Our method yields sparse, time-aware explanations that can be traced across representative frames of the sequence. Our method typically select a much smaller subset of edges per frame, which leads to more visualized frames under the same total edge budget. For pull-ups, the yellow edges focus on the main kinematic chains (e.g., shoulder–elbow–wrist and hip–knee) during both lifting and lowering. For running, they concentrate on the lower limb chain (hip–knee–ankle) that drives forward motion, showing when the discriminative gait pattern appears. For climbing, our method mainly selects the supporting arm and leg that hold the body and control the center of mass, instead of uniformly highlighting the whole skeleton. For sit-ups, the selected edges lie along the torso–hip–knee chain, capturing how the upper body bends around relatively fixed legs.

In contrast, across all four actions, PGExplainer, GNNExplainer, and TGNNExplainer usually produce dense masks on a single frame, where most skeletal links are marked as important, making it hard to see which joints actually drive the prediction. Under a fixed edge budget, this behavior naturally yields fewer visualized frames. TempME often selects edges that are unstable over time and misaligned with the key biomechanics of the movement. In many frames, it highlights secondary body parts and its selected edges do not form a consistent pattern across frames.

## A.9  COMPLEXITY ANALYSIS

**Construct the topology attribution tree**: if $f_{\text{emb}}$ is the graph sum function, the complexity is $O\big(N \cdot d_m + 2^L \cdot B \cdot n \cdot d_m\big)$. If the $f_{\text{emb}}$ is the graph attention function, the complexity is $O\big(N \cdot$

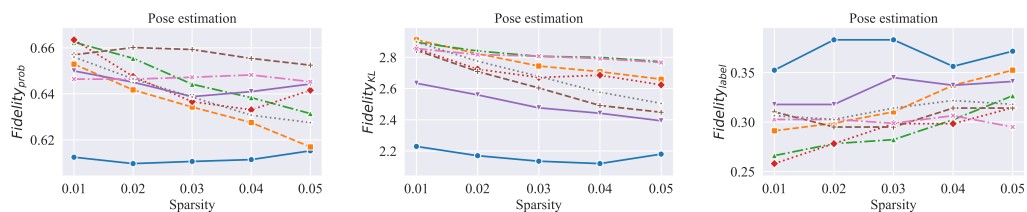

Figure 6: Performance of Fidelity$_{prob}$, Fidelity$_{kl}$ and Fidelity$_{label}$ in pose estimation task.

Table 2: Visualization of explanation results for the pull-up action with different methods. Green nodes denote joint (bone) nodes, red edges denote original connections between joints, yellow edges denote the selected important edges, and each image corresponds to one frame in the video.

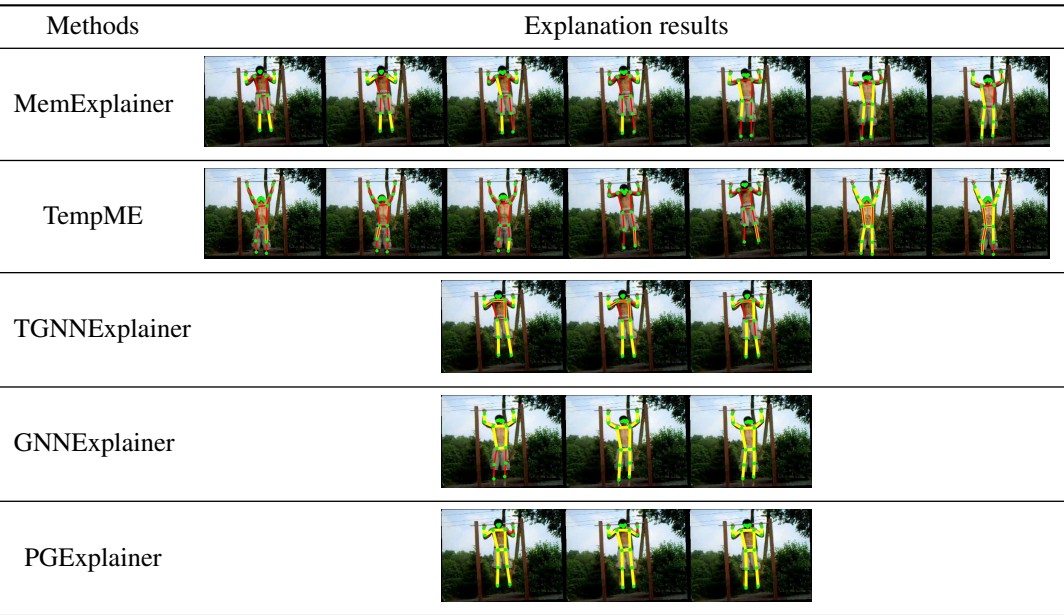

| Methods | Explanation results |
|---|---|
| MemExplainer | |
| TempME | |
| TGNNExplainer | |
| GNNExplainer | |
| PGExplainer | |

Table 3: Visualization of explanation results for the running action with different methods.Green nodes denote joint (bone) nodes, red edges denote original connections between joints, yellow edges denote the selected important edges, and each image corresponds to one frame in the video.

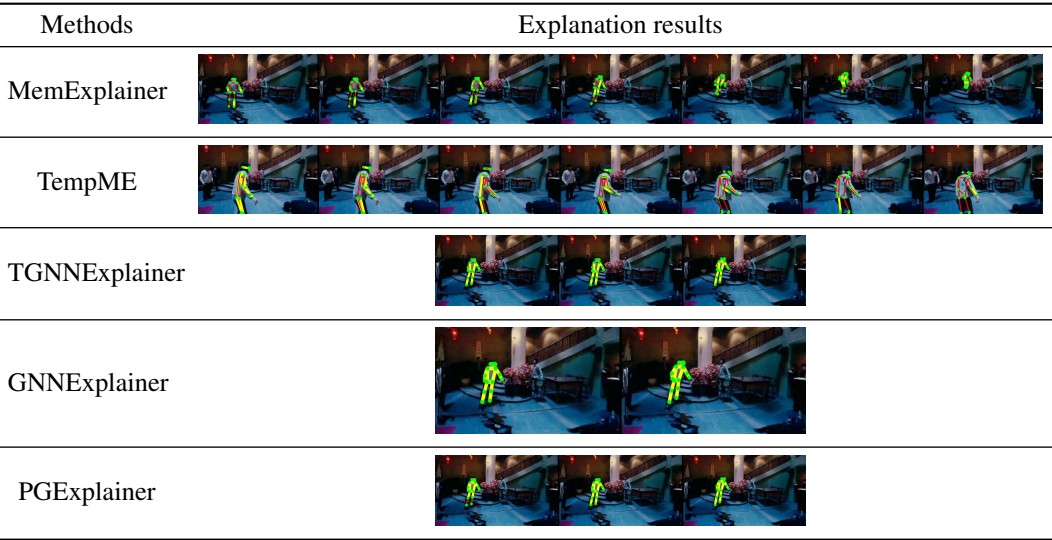

| Methods | Explanation results |
|---|---|
| MemExplainer | |
| TempME | |
| TGNNExplainer | |
| GNNExplainer | |
| PGExplainer | |

Table 4: Visualization of explanation results for the climbing action with different methods.Green nodes denote joint (bone) nodes, red edges denote original connections between joints, yellow edges denote the selected important edges, and each image corresponds to one frame in the video.

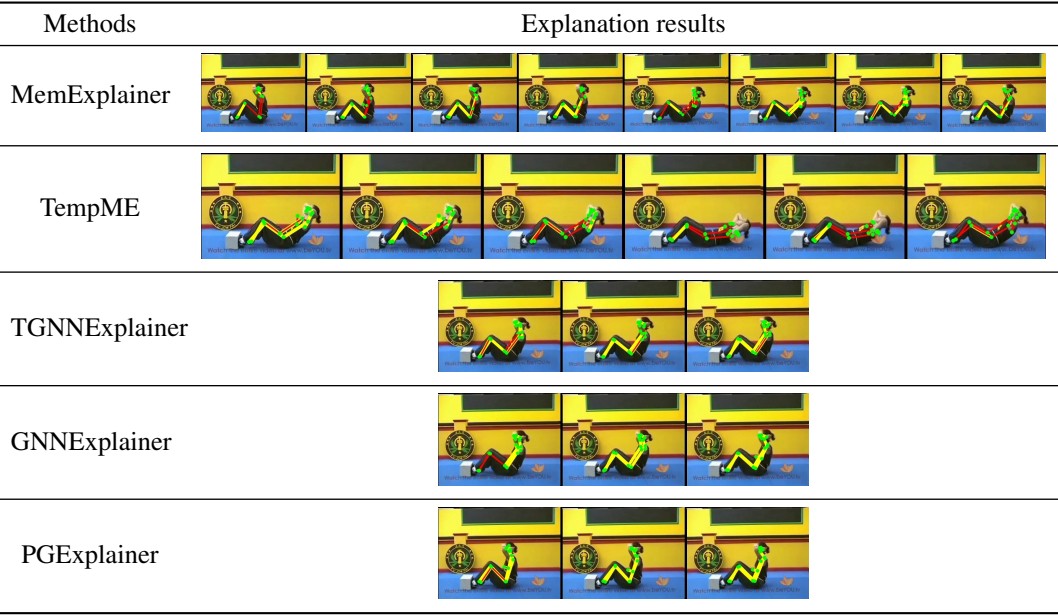

Table 5: Visualization of explanation results for the sit-ups action with different methods.Green nodes denote joint (bone) nodes, red edges denote original connections between joints, yellow edges denote the selected important edges, and each image corresponds to one frame in the video.

$d_m + 2^L \cdot B \cdot (n + n^2) \cdot d_m)$, where $d_m$ is the dimension of the memory vector, $n$ is the number of most recent interactions considered, $L$ is the number of graph layers, $N = |\mathcal{V}(t)|$ is the number of nodes, and $B$ is the batch size.

**Construct the memory backtracking tree**: the complexity is $O(2^T \cdot T \cdot d_m \cdot d_m)$, where $T$ is the depth of memory backtracking tree, and $d_m$ is the dimension of the memory vector.

**Select the important events**: the complexity is $O(d_c^3)$, where $d_c$ is the number of events in final contribution matrix $\mathbf{C}^t$

### A.10 RUNNING TIME

We partition the computation time into three components: **topology time**, for constructing the topology attribution tree and computing neighbor contributions; **memory time**, for constructing the memory backtracking tree and computing contributions of historical events; and **selection time**, for solving Eq. (14). Figure 7 reports the average computation time on all datasets. Topology time is omitted for the enron, UCI, tgbn-trade, tgbn-genre and tgbn-reddit datasets because it accounts for less than 1% of the total runtime and is therefore not shown in the stacked bars.

For link prediction, memory time is the dominant cost. It increases with the memory depth $T$ but remains practical: when $T = 10$, constructing the memory backtracking tree and computing historical contributions takes at most 5 seconds. For node property prediction, selection time dominates because the output is higher-dimensional than in link prediction (where the logit is one-dimensional), making the final solving step more expensive. Even so, when $T = 10$, event selection takes at most 3 seconds, which is still acceptable.

### A.11 THE AVERAGE AND STANDARD DEVIATION OF FIDELITY$_{\text{PROB}}$ AND FIDELITY$_{\text{KL}}$.

Tables 6 and 7 report the mean and standard deviation of Fidelity$_{\text{KL}}$ and Fidelity$_{\text{prob}}$ corresponding to Figures 2 and 3. We further performed a t-test between our method and the second-best baseline, and found statistically significant in 77% of the cases for Fidelity$_{\text{KL}}$ and in 74% of the cases for Fidelity$_{\text{prob}}$.

### A.12 THE PERFORMANCE OF FIDELITY$_{\text{PROB}}$ AND FIDELITY$_{\text{KL}}$ WHEN THE $f_{\text{EMB}}$ IS GRAPH ATTENTION MODEL.

In Figure 8 and Figure 9, we show the performance of Fidelity$_{\text{prob}}$ and Fidelity$_{\text{KL}}$ when the $f_{\text{emb}}$ is graph attention model. Our method, MemExplainer, significantly outperform baseline explainers across all datasets for both metrics.

### A.13 THE PERFORMANCE OF FIDELITY$_{\text{PROB}}$ AND FIDELITY$_{\text{KL}}$ WHEN THE $f_{\text{UPDATER}}$ IS RNN MODEL.

In Figure 10 and Figure 11, we report Fidelity$_{\text{prob}}$ and Fidelity$_{\text{KL}}$ for the setting where the memory updater $f_{\text{update}}$ is implemented as an RNN. Our method, MemExplainer, consistently and significantly outperforms all baseline explainers across all datasets on both metrics.

### A.14 THE PERFORMANCE OF FIDELITY$_{\text{PROB}}$ AND FIDELITY$_{\text{KL}}$ WHEN THE NUMBER OF EVENTS FROM NEIGHBORING SAMPLES, $n$, IS 20.

In Figure 12 and Figure 13, we report Fidelity$_{\text{prob}}$ and Fidelity$_{\text{KL}}$ when the number of events from neighboring samples is fixed to $n = 20$. Our method, MemExplainer, consistently outperforms all baseline methods across all datasets on both metrics.

### A.15 THE ANALYSIS OF REPETITION COUNT

For each target event, we first obtain a set of important explanatory events $\mathcal{E}^*$ using our proposed method. For each event $e = (u, v, t) \in \mathcal{E}^*$, we compute its repetition count $r(e)$ defined as the total number of times the node pair $(u, v)$ has appeared in the dynamic graph before the current prediction

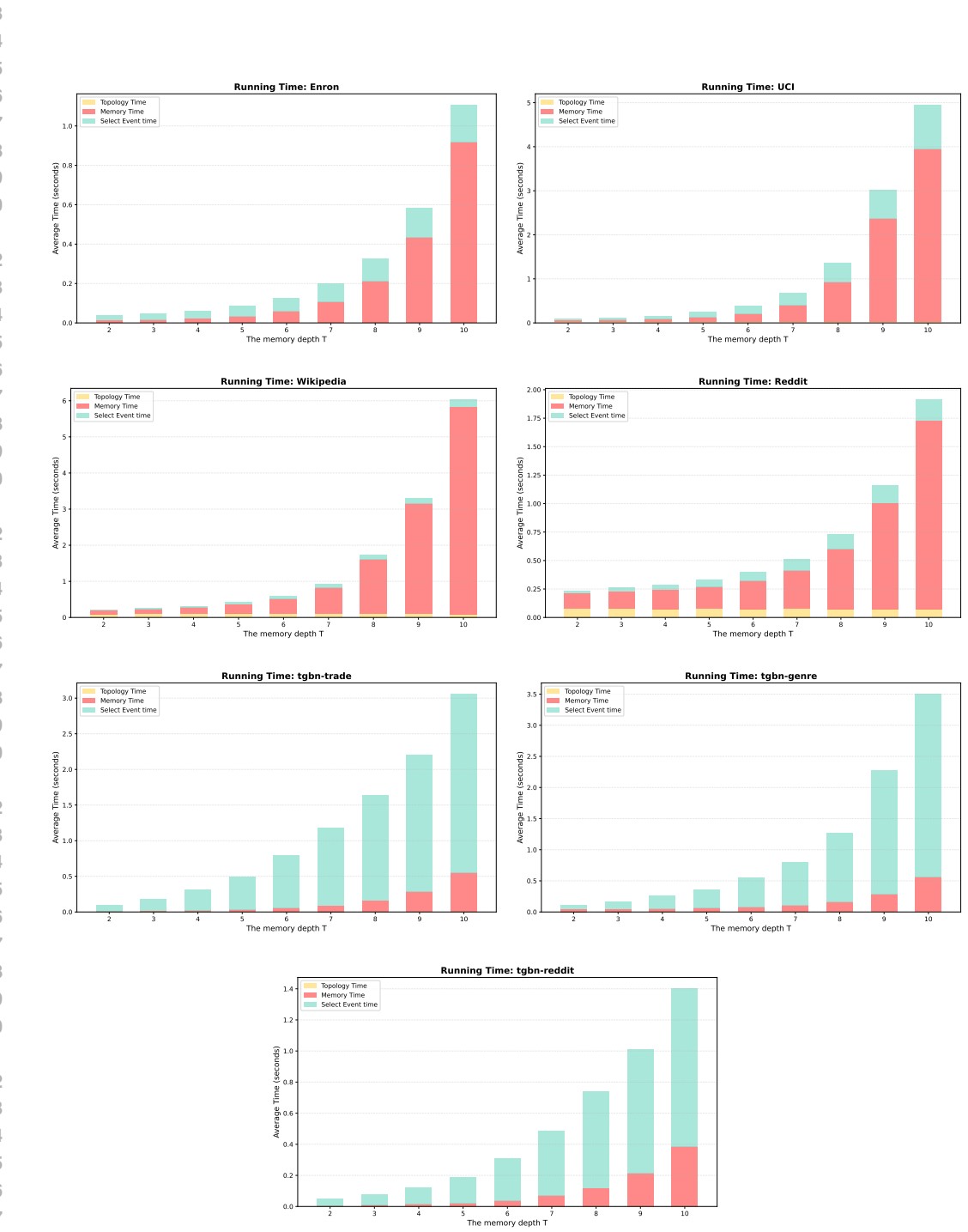

Figure 7: Running time decomposition: each figure represents a dataset.

Table 6: The average and standard deviation of Fidelity$_{KL}$ under different sparsity level. $*$ indicates that our results are significantly different from the runner-up baseline under a t-test.

| Dataset | Ratio | oeov MemExplainer | TempME | TGNNExplainer | GNNExplainer | PGExplainer | w/o memory | w/o topology | w/o selection |
|---|---|---|---|---|---|---|---|---|---|
| Wikipedia | 0.02 | **0.059**±0.113* | 0.103±0.216 | 0.120±0.236 | 0.146±0.294 | 0.110±0.193 | 0.079±0.132 | 0.088±0.162 | 0.104±0.234 |
| | 0.04 | **0.056**±0.104* | 0.095±0.186 | 0.124±0.235 | 0.141±0.280 | 0.105±0.188 | 0.079±0.142 | 0.083±0.168 | 0.098±0.239 |
| | 0.06 | **0.058**±0.114 | 0.090±0.182 | 0.128±0.270 | 0.142±0.307 | 0.105±0.192 | 0.077±0.142 | 0.070±0.142 | 0.094±0.236 |
| | 0.08 | **0.059**±0.116 | 0.081±0.158 | 0.119±0.265 | 0.150±0.313 | 0.107±0.189 | 0.072±0.133 | 0.070±0.143 | 0.091±0.226 |
| | 0.10 | **0.060**±0.119 | 0.077±0.149 | 0.123±0.251 | 0.146±0.337 | 0.120±0.237 | 0.072±0.142 | 0.068±0.140 | 0.088±0.221 |
| Reddit | 0.02 | **0.164**±0.257* | 0.341±0.501 | 0.494±0.553 | 0.436±0.534 | 0.534±0.570 | 0.359±0.439 | 0.583±0.628 | 0.285±0.378 |
| | 0.04 | **0.158**±0.252* | 0.325±0.516 | 0.472±0.574 | 0.428±0.549 | 0.506±0.560 | 0.315±0.414 | 0.801±0.675 | 0.242±0.353 |
| | 0.06 | **0.150**±0.258* | 0.316±0.584 | 0.445±0.555 | 0.418±0.547 | 0.474±0.542 | 0.303±0.407 | 0.942±0.665 | 0.219±0.331 |
| | 0.08 | **0.139**±0.253* | 0.297±0.586 | 0.425±0.529 | 0.391±0.512 | 0.458±0.536 | 0.303±0.446 | 0.942±0.665 | 0.193±0.276 |
| | 0.10 | **0.132**±0.260* | 0.264±0.509 | 0.397±0.480 | 0.387±0.527 | 0.439±0.529 | 0.291±0.410 | 0.942±0.665 | 0.176±0.286 |
| UCI | 0.02 | **0.060**±0.095* | 0.083±0.116 | 0.095±0.135 | 0.095±0.132 | 0.097±0.125 | 0.080±0.102 | 0.071±0.115 | 0.072±0.105 |
| | 0.04 | **0.049**±0.082* | 0.077±0.117 | 0.096±0.141 | 0.097±0.143 | 0.094±0.135 | 0.073±0.098 | 0.075±0.114 | 0.059±0.101 |
| | 0.06 | **0.044**±0.078 | 0.069±0.104 | 0.095±0.146 | 0.095±0.150 | 0.095±0.148 | 0.069±0.095 | 0.084±0.105 | 0.052±0.095 |
| | 0.08 | **0.041**±0.079 | 0.066±0.102 | 0.092±0.143 | 0.094±0.149 | 0.093±0.144 | 0.070±0.096 | 0.089±0.109 | 0.048±0.097 |
| | 0.10 | **0.039**±0.076 | 0.064±0.095 | 0.093±0.145 | 0.092±0.147 | 0.094±0.150 | 0.074±0.096 | 0.095±0.108 | 0.041±0.081 |
| Enron | 0.02 | **0.073**±0.160* | 0.138±0.230 | 0.204±0.218 | 0.187±0.220 | 0.201±0.213 | 0.177±0.225 | 0.120±0.212 | 0.146±0.224 |
| | 0.04 | **0.071**±0.164* | 0.122±0.225 | 0.196±0.225 | 0.183±0.219 | 0.201±0.215 | 0.177±0.227 | 0.101±0.212 | 0.129±0.228 |
| | 0.06 | **0.070**±0.140* | 0.111±0.223 | 0.185±0.225 | 0.183±0.226 | 0.196±0.235 | 0.177±0.245 | 0.085±0.188 | 0.107±0.228 |
| | 0.08 | **0.065**±0.137* | 0.105±0.229 | 0.183±0.228 | 0.181±0.228 | 0.195±0.236 | 0.180±0.245 | 0.078±0.214 | 0.101±0.260 |
| | 0.10 | **0.070**±0.143 | 0.100±0.227 | 0.180±0.230 | 0.179±0.228 | 0.197±0.237 | 0.184±0.251 | 0.076±0.215 | 0.099±0.261 |
| tgbn-trade | 0.02 | **0.153**±0.063 | 0.165±0.052 | 0.203±0.071 | 0.198±0.067 | 0.197±0.065 | 0.155±0.059 | 0.171±0.066 | 0.181±0.063 |
| | 0.04 | **0.143**±0.055* | 0.161±0.049 | 0.203±0.071 | 0.198±0.067 | 0.197±0.065 | 0.148±0.055 | 0.165±0.065 | 0.177±0.063 |
| | 0.06 | **0.137**±0.053* | 0.158±0.046 | 0.203±0.071 | 0.197±0.066 | 0.196±0.064 | 0.142±0.055 | 0.158±0.060 | 0.173±0.061 |
| | 0.08 | **0.130**±0.048* | 0.155±0.044 | 0.203±0.071 | 0.196±0.064 | 0.195±0.062 | 0.137±0.056 | 0.151±0.055 | 0.169±0.061 |
| | 0.10 | **0.125**±0.046* | 0.152±0.044 | 0.203±0.071 | 0.196±0.063 | 0.193±0.059 | 0.131±0.057 | 0.144±0.055 | 0.166±0.061 |
| tgbn-genre | 0.02 | **0.638**±0.133* | 0.705±0.095 | 0.745±0.103 | 0.751±0.099 | 0.760±0.119 | 0.714±0.122 | 0.671±0.114 | 0.729±0.118 |
| | 0.04 | **0.614**±0.132* | 0.687±0.101 | 0.751±0.103 | 0.751±0.097 | 0.754±0.119 | 0.709±0.123 | 0.656±0.120 | 0.721±0.124 |
| | 0.06 | **0.604**±0.133* | 0.678±0.104 | 0.744±0.104 | 0.750±0.096 | 0.752±0.117 | 0.706±0.125 | 0.645±0.121 | 0.705±0.129 |
| | 0.08 | **0.589**±0.128* | 0.666±0.107 | 0.738±0.107 | 0.749±0.094 | 0.752±0.117 | 0.694±0.126 | 0.628±0.121 | 0.700±0.129 |
| | 0.10 | **0.577**±0.125* | 0.659±0.106 | 0.735±0.107 | 0.748±0.094 | 0.755±0.119 | 0.690±0.127 | 0.618±0.118 | 0.694±0.130 |
| tgbn-reddit | 0.02 | **0.682**±0.414* | 1.090±0.727 | 1.732±0.686 | 1.535±0.674 | 1.872±0.769 | 0.850±0.676 | 0.858±0.610 | 0.979±0.525 |
| | 0.04 | **0.565**±0.368* | 0.967±0.714 | 1.722±0.721 | 1.693±0.721 | 1.871±0.766 | 0.773±0.659 | 0.734±0.577 | 0.912±0.491 |
| | 0.06 | **0.470**±0.325* | 0.868±0.703 | 1.768±0.729 | 1.708±0.719 | 1.869±0.761 | 0.719±0.663 | 0.635±0.536 | 0.835±0.454 |
| | 0.08 | **0.406**±0.291* | 0.788±0.695 | 1.825±0.755 | 1.722±0.722 | 1.834±0.745 | 0.672±0.651 | 0.548±0.504 | 0.799±0.447 |
| | 0.10 | **0.341**±0.254* | 0.726±0.693 | 1.766±0.773 | 1.767±0.727 | 1.806±0.731 | 0.647±0.646 | 0.477±0.457 | 0.748±0.434 |

Table 7: The average and standard deviation of Fidelity$_{prob}$ under different sparsity level. $*$ indicates that our results are significantly different from the runner-up baseline under a t-test.

| Dataset | Ratio | MemExplainer | TempME | TGNNExplainer | GNNExplainer | PGExplainer | w/o memory | w/o topology | w/o selection |
|---|---|---|---|---|---|---|---|---|---|
| Wikipedia | 0.02 | **0.085**±0.108* | 0.122±0.144 | 0.131±0.150 | 0.144±0.164 | 0.129±0.142 | 0.106±0.114 | 0.110±0.127 | 0.116±0.147 |
| | 0.04 | **0.084**±0.105* | 0.118±0.140 | 0.133±0.149 | 0.141±0.157 | 0.123±0.139 | 0.104±0.112 | 0.105±0.128 | 0.108±0.144 |
| | 0.06 | **0.083**±0.106 | 0.113±0.136 | 0.130±0.153 | 0.138±0.160 | 0.121±0.140 | 0.101±0.114 | 0.094±0.119 | 0.106±0.141 |
| | 0.08 | **0.083**±0.106 | 0.107±0.129 | 0.123±0.147 | 0.139±0.164 | 0.123±0.141 | 0.098±0.108 | 0.091±0.122 | 0.103±0.139 |
| | 0.10 | **0.084**±0.108 | 0.103±0.125 | 0.124±0.147 | 0.136±0.157 | 0.126±0.147 | 0.096±0.109 | 0.090±0.121 | 0.100±0.137 |
| Reddit | 0.02 | **0.181**±0.173* | 0.279±0.228 | 0.354±0.249 | 0.321±0.249 | 0.373±0.256 | 0.293±0.242 | 0.379±0.285 | 0.260±0.215 |
| | 0.04 | **0.175**±0.171* | 0.264±0.229 | 0.335±0.252 | 0.312±0.252 | 0.357±0.255 | 0.268±0.231 | 0.474±0.300 | 0.232±0.203 |
| | 0.06 | **0.169**±0.167* | 0.247±0.228 | 0.320±0.254 | 0.305±0.250 | 0.339±0.256 | 0.256±0.231 | 0.539±0.285 | 0.218±0.192 |
| | 0.08 | **0.159**±0.162* | 0.232±0.224 | 0.310±0.253 | 0.292±0.248 | 0.330±0.256 | 0.248±0.234 | 0.539±0.285 | 0.204±0.183 |
| | 0.10 | **0.152**±0.158* | 0.219±0.212 | 0.299±0.245 | 0.287±0.249 | 0.319±0.255 | 0.245±0.233 | 0.539±0.285 | 0.188±0.181 |
| Enron | 0.02 | **0.115**±0.127* | 0.192±0.133 | 0.252±0.141 | 0.234±0.144 | 0.250±0.141 | 0.226±0.151 | 0.180±0.121 | 0.203±0.133 |
| | 0.04 | **0.111**±0.124* | 0.173±0.134 | 0.242±0.146 | 0.229±0.146 | 0.248±0.145 | 0.223±0.154 | 0.151±0.123 | 0.178±0.138 |
| | 0.06 | **0.117**±0.117* | 0.159±0.131 | 0.231±0.147 | 0.226±0.152 | 0.240±0.149 | 0.219±0.159 | 0.135±0.116 | 0.150±0.137 |
| | 0.08 | **0.104**±0.118* | 0.149±0.133 | 0.226±0.153 | 0.223±0.154 | 0.239±0.150 | 0.220±0.161 | 0.119±0.121 | 0.134±0.142 |
| | 0.10 | **0.111**±0.121 | 0.141±0.134 | 0.220±0.156 | 0.220±0.156 | 0.238±0.154 | 0.218±0.169 | 0.116±0.124 | 0.128±0.142 |
| UCI | 0.02 | **0.100**±0.101* | 0.127±0.116 | 0.135±0.121 | 0.136±0.122 | 0.140±0.120 | 0.125±0.104 | 0.111±0.112 | 0.115±0.108 |
| | 0.04 | **0.086**±0.091* | 0.119±0.117 | 0.133±0.126 | 0.133±0.128 | 0.133±0.123 | 0.115±0.102 | 0.117±0.114 | 0.096±0.103 |
| | 0.06 | **0.081**±0.087 | 0.109±0.112 | 0.129±0.127 | 0.128±0.129 | 0.129±0.127 | 0.110±0.101 | 0.130±0.113 | 0.087±0.097 |
| | 0.08 | **0.075**±0.085 | 0.105±0.111 | 0.125±0.125 | 0.126±0.129 | 0.126±0.127 | 0.110±0.102 | 0.137±0.114 | 0.079±0.096 |
| | 0.10 | **0.073**±0.083 | 0.103±0.109 | 0.126±0.127 | 0.124±0.127 | 0.125±0.128 | 0.117±0.105 | 0.145±0.111 | 0.073±0.087 |
| tgbn-trade | 0.02 | **0.072**±0.042 | 0.074±0.043 | 0.077±0.047 | 0.076±0.046 | 0.076±0.045 | 0.073±0.043 | 0.075±0.042 | 0.077±0.045 |
| | 0.04 | **0.070**±0.038* | 0.073±0.042 | 0.077±0.047 | 0.076±0.046 | 0.076±0.045 | 0.072±0.043 | 0.074±0.041 | 0.076±0.045 |
| | 0.06 | **0.069**±0.037* | 0.072±0.041 | 0.077±0.047 | 0.076±0.045 | 0.076±0.045 | 0.071±0.043 | 0.072±0.040 | 0.076±0.044 |
| | 0.08 | **0.067**±0.035* | 0.072±0.040 | 0.077±0.047 | 0.076±0.045 | 0.076±0.044 | 0.070±0.043 | 0.071±0.037 | 0.075±0.044 |
| | 0.10 | **0.066**±0.034* | 0.071±0.040 | 0.077±0.047 | 0.076±0.045 | 0.076±0.044 | 0.069±0.043 | 0.070±0.037 | 0.075±0.044 |
| tgbn-genre | 0.02 | **0.115**±0.017* | 0.118±0.017 | 0.119±0.017 | 0.119±0.017 | 0.120±0.018 | 0.118±0.018 | 0.117±0.017 | 0.119±0.018 |
| | 0.04 | **0.114**±0.017* | 0.117±0.016 | 0.119±0.017 | 0.119±0.017 | 0.119±0.018 | 0.118±0.018 | 0.116±0.018 | 0.118±0.018 |
| | 0.06 | **0.113**±0.017* | 0.117±0.016 | 0.119±0.017 | 0.119±0.017 | 0.119±0.018 | 0.118±0.018 | 0.115±0.018 | 0.118±0.018 |
| | 0.08 | **0.113**±0.016* | 0.116±0.016 | 0.119±0.017 | 0.119±0.017 | 0.119±0.018 | 0.117±0.018 | 0.115±0.018 | 0.118±0.018 |
| | 0.10 | **0.112**±0.016* | 0.116±0.016 | 0.118±0.017 | 0.119±0.017 | 0.119±0.018 | 0.117±0.018 | 0.114±0.018 | 0.117±0.018 |
| tgbn-reddit | 0.02 | **0.245**±0.088 | 0.295±0.108 | 0.374±0.073 | 0.357±0.078 | 0.380±0.074 | 0.255±0.114 | 0.265±0.103 | 0.296±0.085 |
| | 0.04 | **0.221**±0.086* | 0.276±0.113 | 0.371±0.076 | 0.368±0.078 | 0.380±0.073 | 0.239±0.118 | 0.243±0.103 | 0.287±0.084 |
| | 0.06 | **0.199**±0.082* | 0.257±0.116 | 0.374±0.076 | 0.370±0.077 | 0.380±0.074 | 0.227±0.121 | 0.224±0.103 | 0.276±0.084 |
| | 0.08 | **0.183**±0.078* | 0.241±0.120 | 0.377±0.075 | 0.371±0.076 | 0.379±0.074 | 0.216±0.123 | 0.204±0.103 | 0.270±0.085 |
| | 0.10 | **0.165**±0.073* | 0.226±0.123 | 0.371±0.078 | 0.375±0.075 | 0.377±0.074 | 0.210±0.123 | 0.189±0.100 | 0.261±0.085 |

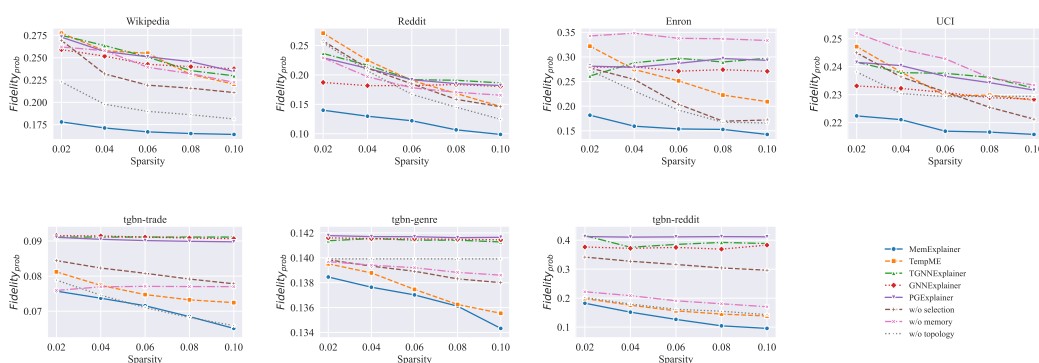

Figure 8: The performance of Fidelity$_{\text{KL}}$. Each figure corresponds to a different dataset. First and second rows represent link prediction and node property prediction, respectively. Lower value indicates better performance. The $f_{\text{emb}}$ is graph attention model.

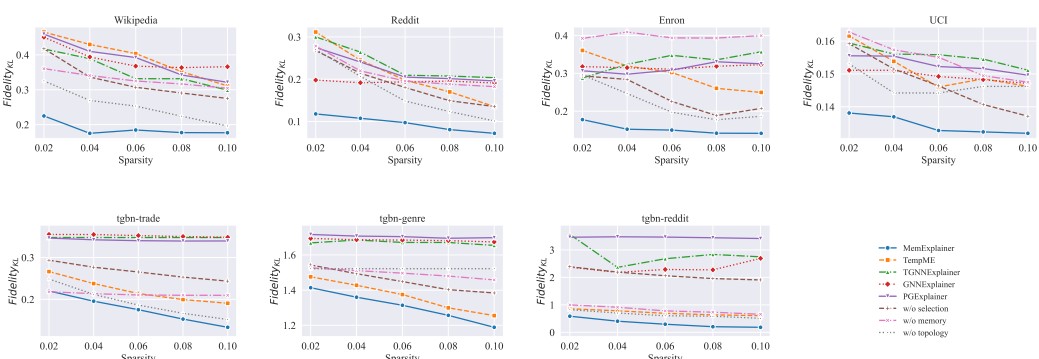

Figure 9: The performance of Fidelity$_{\text{prob}}$. Each figure corresponds to a different dataset. First and second rows represent link prediction and node property prediction, respectively. Lower value indicates better performance. The $f_{\text{emb}}$ is graph attention model.

time. We then calculate the average repetition of the explanatory set as $\bar{r} = \frac{\sum_{e \in \mathcal{E}^*} r(e)}{|\mathcal{E}^*|}$. On the Wikipedia, UCI, and Reddit datasets, we compute $\bar{r}$ for 500 target events and plot the corresponding histograms in Figure 14. On the UCI dataset, about 50% of target events have $\bar{r}$ between 1 and 5.75. On the Wikipedia dataset, about 28% of target events have $\bar{r}$ between 1 and 10. On the Reddit dataset, about 60% of target events have $\bar{r}$ between 1 and 3. Thus, on Wikipedia, important events often use highly repeated edges, while on Reddit, the important events involve rarely repeated edges.

## A.16 THE DERIVATION OF EQUATION (14)

In Eq. (13), we derived the KL divergence between $p_2$ and $p_1$, where $p_1$ and $p_2$ are arbitrary probabilities obtained by applying the sigmoid function to logits. Now we let $p_2$ be the predicted probability on the original dynamic graph, $p_2 = \hat{y}_{v_k, u_k}(\mathcal{E}(t)) = \sigma(z_2)$, and $p_1$ be the predicted probability after retaining only a subset of critical events and removing the others. Based on the event-contribution matrix $\mathbf{C}^t$, we approximate the perturbed logit $z_1$ as, $z_1 = \sum_{l=1}^{d_c} d_l \mathbf{C}^t_{e_l}$, where $d_c$ is the number of candidate events, and $\mathbf{d} \in \{0, 1\}^{d_c}$ is a selection vector whose entry $d_l$ indicates whether event $e_l$ is

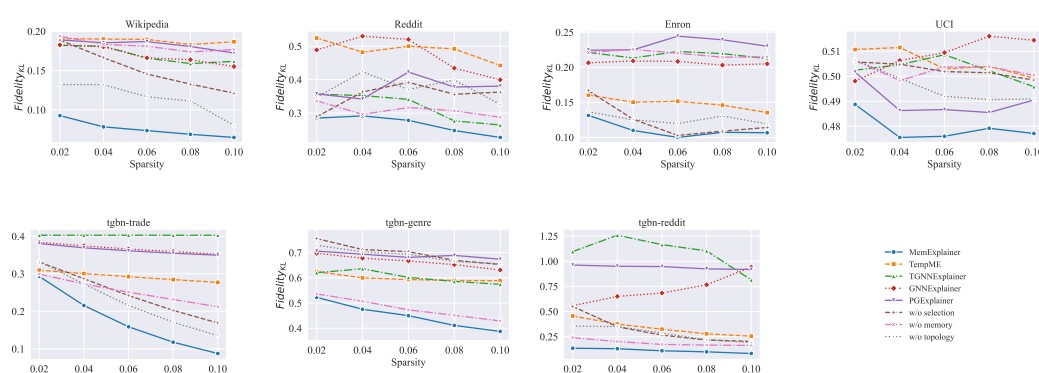

Figure 10: The performance of Fidelity$_{\text{KL}}$. Each figure corresponds to a different dataset. First and second rows represent link prediction and node property prediction, respectively. Lower value indicates better performance. The $f_{\text{update}}$ is rnn model.

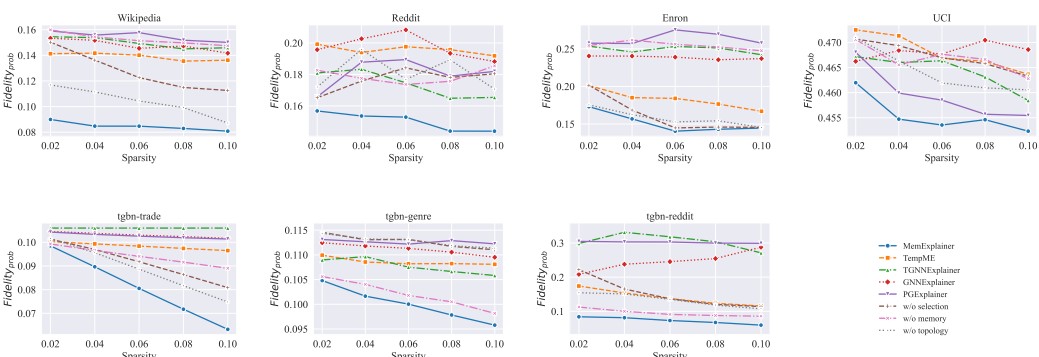

Figure 11: The performance of Fidelity$_{\text{prob}}$. Each figure corresponds to a different dataset. First and second rows represent link prediction and node property prediction, respectively. Lower value indicates better performance. The $f_{\text{update}}$ is rnn model.

selected. Substituting $z_1$ into Eq. (13) gives

$$\text{KL}(p_2 \parallel p_1) = \sigma(z_2)(z_2 - z_1) - \text{sp}(z_2) + \text{sp}(z_1)$$

$$= \sigma(z_2)z_2 - \hat{y}_{v_k, u_k}\big(\mathcal{E}(t)\big) \sum_{l=1}^{d_c} d_l \mathbf{C}_{e_l}^t - \text{sp}(z_2) + \text{sp}(\sum_{l=1}^{d_c} d_l \mathbf{C}_{e_l}^t)$$

Since $\sigma(z_2)z_2$ and $\text{sp}(z_2)$ do not depend on $\mathbf{d}$, they are constant with respect to the selection and can be dropped from the optimization objective. Minimizing the remaining KL term over $\mathbf{d}$ yields Eq. (14), which we use to select a small set of key events that best preserves the original prediction.

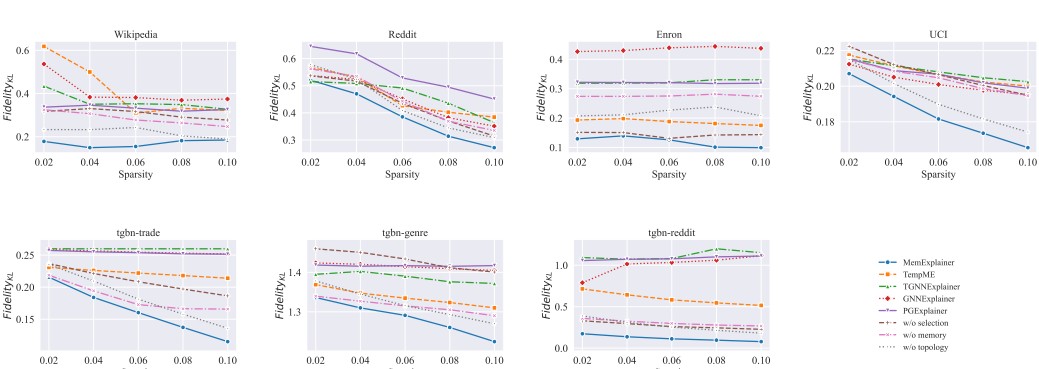

Figure 12: The performance of Fidelity$_{KL}$. Each figure corresponds to a different dataset. First and second rows represent link prediction and node property prediction, respectively. Lower value indicates better performance. The number of events from neighboring samples $n$ is 20.

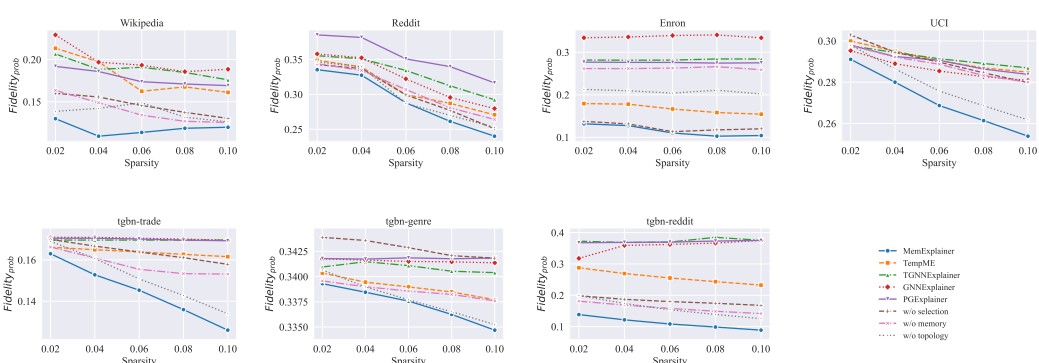

Figure 13: The performance of Fidelity$_{prob}$. Each figure corresponds to a different dataset. First and second rows represent link prediction and node property prediction, respectively. Lower value indicates better performance. The number of events from neighboring samples $n$ is 20.

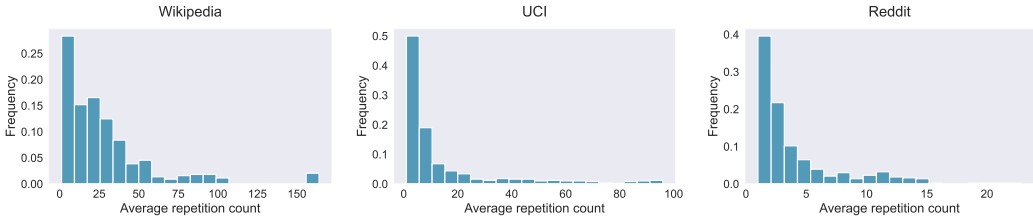

Figure 14: The histograms of average repetition in Wikipedia, UCI and Reddit datasets.

