# OpenReview forum: "Towards the Explainability of Temporal Graph Networks via Memory Backtracking"
_ICLR.cc/2026/Conference — Submitted to ICLR 2026_

### Official Review · Reviewer_6Mxv · 2025-10-21

**Soundness:** 2
**Presentation:** 2
**Contribution:** 2
**Rating:** 2
**Confidence:** 3

**Summary:**

The paper studies explainability for Temporal Graph Networks. It introduces MemExplainer that attributes a target prediction to recent interactions and to stored node memories. The approach builds a topology attribution tree and then backtracks memory updates to earlier events. The authors claim a conservation-style property linking the final logit to the sum of selected event contributions. Experiments are shown across several temporal-graph tasks and datasets using figures to suggest improved fidelity.

**Strengths:**

1. The work targets a real need for explanations that account for temporal memory in graph models.

2. The pipeline is clear at a high level, with a split between topology attribution and memory backtracking and an emphasis on contribution conservation.

3. The evaluation spans multiple datasets and both link prediction and node-property prediction, which increases practical relevance.

**Weaknesses:**

1. The evaluation relies on model-internal fidelity measures that compare outputs before and after event selection. These measures can reward matching model behavior rather than faithfulness to the underlying phenomenon. There is no human study, no causal validation, and no task-utility assessment that would confirm usefulness beyond reproducing the model.

2. Key ablations and robustness checks are missing. The paper does not ablate the memory-backtracking component or the topology-attribution component, so their individual contributions are not isolated. And there is little analysis of sensitivity to timestamp noise, to alternative memory-update modules, or to choices that control recent-event sampling and backtracking depth. Multi-seed statistics and significance testing are not reported.

3. Computational cost is not reported. The method builds and traverses attribution structures over time, yet there are no numbers for run-time or peak-memory, or how these scale with graph size, event density, number of layers, or backtracking depth.

4. Baseline and metric coverage could be stronger. Only a small set of explainers is considered, and some are adapted from static-graph settings.

5. Lack of reproducibility details. Key implementation choices are not clearly described, including how node memories are built and updated, when event selection stops, and what optimizer is used to pick the event set. The paper does not mention code release or random seeds.

**Questions:**

1. What are the typical run-time and memory costs per instance, and how do they change with graph size, number of sampled recent events, number of layers, and backtracking depth?

2. Can you provide ablations that remove the memory-backtracking stage, remove the topology-attribution stage, and replace the event-selection objective with a simple top-$k$ rule to quantify each component’s contribution?

3. How sensitive are the explanations to timestamp perturbations, to noisy features, and to alternative memory-update modules? Do the explanations transfer across different TGN variants?

4. Do you have validation beyond model matching, such as human judgments or interventional tests that remove or inject events to confirm that selected events are truly explanatory?

5. How is the optimization over the selected event set solved in practice?

---

> ### Author Response · Authors · 2025-11-26
> **Response to Reviewer 6Mxv (1)**
>
> Thank you for taking the time to review our paper!
>
> >**Q1: There is no human study, no causal validation, and no task-utility assessment that would confirm usefulness beyond reproducing the model.**
>
> **A1**:  We further evaluate our method on a **human pose estimation** task and present case studies that illustrate the interpretability of the resulting explanations. For each video, we first apply YOLOPose to detect human keypoints and construct a skeleton graph whose nodes are body joints and whose edges follow the human kinematic structure. This yields a sequence of skeleton graphs (one per frame), which we feed into TGN to predict the action label. We use the HMDB51 dataset, and select four intuitive classes, sit-up, pull-up, climbing, and running, as our training and evaluation subset.
>
> For a fair comparison, all explanation methods operate on exactly the same input sequence of skeleton graphs for each video. We also fix a **global edge budget**, i.e., the total number of key human-pose edges that each explanation method is allowed to select for that video. In addition to $Fidelity_{prob}$ and $Fidelity_{kl}$, we introduce $Fidelity_{label}$, which is set to 1 if the label predicted by TGN **using only the selected important edges** matches the true label, and 0 otherwise; we then average this value over all videos. **Figure 6 in appendix** reports the performance of $Fidelity_{prob}$,  $Fidelity_{kl}$, and $Fidelity_{label}$. **Higher** $Fidelity_{label}$ and **lower** $Fidelity_{prob}$ and $Fidelity_{kl}$ indicate better explanations. As shown in Figure 6, **our method consistently outperforms all baselines on this pose estimation task**.
>
> In **Tables 2, 3, 4, and 5**, we present case studies for the **pull-up**, **running**, **climbing**, and **sit-up** classes. Green nodes denote joint (bone) nodes, red edges denote the original skeletal connections, yellow edges denote the selected important edges, and each image corresponds to one frame from the video.
>
> Our method yields **sparse, time-aware** explanations that can be traced across representative frames of the sequence and are aligned with human biomechanical intuition. Our method typically select a **much smaller subset of edges** per frame, which leads to **more selected (visualized) frames** under the same total edge budget.  For **pull-ups**, the yellow edges concentrate on the **main kinematic chains** (e.g., shoulder–elbow–wrist and hip–knee) during both the lifting and lowering phases. For **running**, they focus on the **lower-limb chain (hip–knee–ankle)** that drives forward motion, making it clear when the discriminative gait pattern appears. For **climbing**, our method mainly selects the **supporting arm and leg** that hold the body and control the center of mass, rather than uniformly highlighting the entire skeleton.
> For **sit-ups**, the selected edges follow the **torso–hip–knee** chain, capturing how the upper body bends around relatively fixed legs.
>
> In contrast, across all four actions, **PGExplainer**, **GNNExplainer**, and **TGNNExplainer** typically produce **dense masks on a single frame, marking most skeletal links as important** and making it difficult to identify which joints actually drive the prediction.  Under a fixed edge budget, this behavior naturally yields **fewer selected (visualized)** frames. **TempME** often selects edges that fluctuate over time and are misaligned with the key biomechanics of the movement: in many frames, it highlights secondary body parts, and **its selected edges do not form a consistent pattern across frames**.
>
> >**Q2: Key ablations and robustness checks are missing. The paper does not ablate the memory-backtracking component or the topology-attribution component, so their individual contributions are not isolated. Statistics and significance testing are not reported.**
>
> **A2**: Thanks for this good suggestion. We add there ablation variants as baselines and update the results in **Figure 2** and **Figure 3**. **We include these three methods as baselines in all experiments**.
>
> **w/o memory** removes the memory backtracking tree and selects events solely based on the contributions of neighboring events.
>
> **w/o topology** removes the topology attribution tree and selects events solely based on the contributions of historical events.
>
> **w/o selection** does not use the objective function to select events; instead, it accumulates event contributions and chooses the final explanatory events using a top-k rule. As shown in **Figure 2** and **Figure 3**, our method consistently outperforms all baselines across all datasets. **Tables 6 and 7 in the appendix** report the mean and standard deviation of $Fidelity_{KL}$ and $Fidelity_{prob}$ corresponding to **Figures 2 and 3**. We further performed a t-test between our method and the second-best baseline, and found statistically significant differences in **77%** of the cases for $Fidelity_{KL}$ and in **74%** of the cases for $Fidelity_{prob}$.

---

> ### Author Response · Authors · 2025-11-26
> **Response to Reviewer 6Mxv (2)**
>
> >**Q3: Computational cost is not reported. What are the typical run-time and memory costs per instance**
>
> **A3**: We provide the complexity analysis in **Appendix A.9**.
>
> **Construct the topology attribution tree**: if $f_{\textnormal{emb}}$ is the graph sum function, the complexity is $O\big(N \cdot d_m+2^L\cdot B\cdot n \cdot d_m)$. If the $f_{\textnormal{emb}}$ is the graph attention function, the complexity is $O\big(N \cdot d_m+2^L\cdot B\cdot (n+n^2) \cdot d_m)$, where $d_m$ is the dimension of the memory vector, $n$ is the number of most recent interactions considered, $L$ is the number of graph layers, $N = |\mathcal{V}(t)|$ is the number of nodes, and $B$ is the batch size.
>
> **Construct the memory backtracking tree**:  the complexity is $O\big(2^T\cdot T\cdot d_m\cdot d_m)$, where $T$ is the depth of memory backtracking tree, and $d_m$ is the dimension of the memory vector.
>
> **Select the important events**: the complexity is $O\big(d_c^3)$, where $d_c$ is the number of events in final contribution matrix $\mathbf{C}^t$
>
> Besides, in **Figure 7** in appendix, we report the average per-instance running time for different depths T across all datasets. We partition the computation time into three components: **topology time**, for constructing the topology attribution tree and computing neighbor contributions; **memory time**, for constructing the memory backtracking tree and computing contributions of historical events; and **selection time**, for solving Eq. (14). Topology time is omitted for the enron, UCI, tgbn-trade, tgbn-genre and tgbn-reddit datasets because it accounts for less than 1% of the total runtime and is therefore not shown in the stacked bars. **For link prediction, memory time is the dominant cost**. It increases with the memory depth $T$ but remains practical: when $T = 10$, constructing the memory backtracking tree and computing historical contributions takes **at most 5 seconds**.  **For node property prediction, selection time dominates cost** because the output is higher-dimensional than in link prediction (where the logit is one-dimensional), making the final solving step more expensive. Even so, when $T = 10$, **event selection takes at most 3 seconds**, which is still acceptable.
>
> >**Q4:And there is little analysis of sensitivity to timestamp noise, to alternative memory-update modules, or to choices that control recent-event sampling and backtracking depth.**
>
> **A4**:  Our data contain real timestamps, so there is no timestamp noise.
>
> For **alternative memory-update modules**, we evaluate the case where the memory updater is **RNN**. In this setting, we apply LRP to the RNN-based memory update function and then use the same Memory Backtracking Tree to trace relevance back to historical events. **Figures 10 and 11** show that, under these RNN updaters, our method, MemExplainer, again outperforms the baselines on all datasets and metrics.
>
> For the **recent-event sampling** setting, **Figures 2 and 3** report results when we sample the **10** most recent events. We also evaluate sampling **20** recent events. **Figures 12 and 13** show that, under this setting, MemExplainer outperforms the baselines on all datasets and metrics.
>
> For the **backtracking depth**, on the Enron, Reddit, UCI, and tgbn-genre datasets we vary the maximum depth of the Memory Backtracking Tree from **2 to 10**, while selecting the same number of events for each target node or edge across all depths. The performance of $Fidelity_{KL}$ and $Fidelity_{prob}$ is shown in **Figures 4 and 5**, respectively. As the maximum depth increases, $Fidelity_{KL}$ and $Fidelity_{prob}$ **first decrease and then increase**. This pattern arises because deeper backtracking enables more accurate tracing of historical events, leading to more faithful explanations, but also enlarges the candidate set of explanatory events, which makes the optimization problem harder and can initially reduce fidelity. The optimal depth differs across datasets, reflecting their distinct characteristics. **Except for the experiments in Figures 4 and 5, we fix the maximum depth to 5 in all other results**.

---

> ### Author Response · Authors · 2025-11-26
> **Response to Reviewer 6Mxv (3)**
>
> >**Q5: Do the explanations transfer across different TGN variants?**
>
> **A5**: More generally, our framework can be extended to other TGNs **as long as the TGN’s memory update function and embedding/message function can be decomposed using LRP**, i.e., LRP can be used to assign contribution scores to their inputs. In such cases, the same Topology Attribution Tree and Memory Backtracking Tree proposed in our work can be applied to propagate and aggregate contributions over neighboring and historical events, followed by the same selection procedure.
>
> For **attention-based** updaters, we extend the LRP allocation rule to the graph attention mechanism.We follow the LRP formulation for graph attention networks in prior work [1] to propagate relevance through attention weights and message transformations. After that, we reuse the same Topology Attribution Tree described in the paper to aggregate the contributions of neighboring events over time. The corresponding experimental results are reported in **Figures 8 and 9**, **where our method, MemExplainer consistently outperforms baseline explainers across all datasets and both evaluation metrics.**
>
> We also test the performance of **RNN** updaters, we similarly apply LRP to the RNN memory update function and then use the same Memory Backtracking Tree to trace relevance back to historical events. Figures **10 and 11** show that, under these RNN updaters, **our method, MemExplainer again achieves better performance than the baselines on all datasets and metrics**.
>
> [1] Attnlrp: attention-aware layer-wise relevance propagation for transformers.
>
> >**Q6: How is the optimization over the selected event set solved in practice?**
>
>  **A6**: Eq. (14) can be solved in two ways: (1) **exact optimization**, where the selection variables are constrained to be binary (0/1) and the final explanatory events are directly determined by the variables equal to 1; and (2) **relaxed optimization**, where the selection variables are allowed to lie in [0,1] and the final explanatory events are obtained by taking the top-K variables with the largest values. We implement Eq. (14) in Python using the **CVXPY** library with **MOSEK** as the solver. All results reported in the paper are based on the **exact (binary) formulation**. In  **Figure 7** in appendix, we show the selection time for solving Eq. (14) across memory depths T and datasets. When $T = 10$, event selection takes at most 3 seconds, which is still acceptable.
>
> >**Q7: The evaluation relies on model-internal fidelity measures that compare outputs before and after event selection. These measures can reward matching model behavior rather than faithfulness to the underlying phenomenon. Do you have validation beyond model matching, such as human judgments or interventional tests that remove or inject events to confirm that selected events are truly explanatory?**
>
> **A7**: We thank the reviewer for raising this point. In practice, quantitative evaluation of explainability methods **almost always relies on fidelity metrics** [2], which measure how well an explanation reproduces the behavior of a trained model. Directly evaluating faithfulness to the underlying phenomenon is **extremely challenging**, because **people not have ground-truth explanations**, and large-scale human evaluation of whether explanations are “intuitive” is typically very time-consuming and costly. In our view, faithfulness to the phenomenon is therefore necessarily mediated by the model. The model must first learn the relevant structure in the data, and an explanation is then judged by how accurately it recovers the internal evidence that the model itself uses. Under this standard assumption, high model fidelity is interpreted as a proxy for faithfulness to the phenomenon.
>
> In our work, we use two metrics, $Fidelity_{KL}$ and $Fidelity_{prob}$, defined in an interventional way. For each target prediction, we first compute the model’s output on the full temporal graph, and then recompute it when only the selected explanatory events are retained, i.e., all non-selected events are removed.**The smaller the change in predicted probability or KL divergence, the better the explanation**. Because it shows that the selected events alone are sufficient to support the original prediction. In the additional pose estimation experiment, we further introduce $Fidelity_{label}$​, which measures whether the label predicted from **only the selected important edges match the ground-truth label (higher is better)**. Thus, all of our quantitative evaluations are explicitly based on interventional tests, only keeping events/edges. The results consistently show that our method yields more faithful explanations than the baselines.
>
> [2] Explainability in graph neural networks: A taxonomic survey

---

> ### Author Response · Authors · 2025-11-26
> **Response to Reviewer 6Mxv (4)**
>
> >**Q8: Baseline and metric coverage could be stronger. Only a small set of explainers is considered, and some are adapted from static-graph settings.**
>
> **A8**: Currently, research on temporal graph explainability is still relatively limited. Only a few methods exist, among which TGNNExplainer and TempME are the most competitive and widely used. We therefore adopt these two methods as our main baselines, and they are close to the state of the art for explainability in TGNs. In addition, we include static-graph explainers because both the TGNNExplainer and TempME papers compare against static-graph methods. Following this setup, we also select several representative explainers for static GNNs for comparison.
>
> >**Q9: Lack of reproducibility details. Key implementation choices are not clearly described, including how node memories are built and updated, when event selection stops, and what optimizer is used to pick the event set. The paper does not mention code release or random seeds.**
>
> **A9**: **Node memories**. For all experiments, each node’s memory vector is initialized to the zero vector at the beginning of training. During training and evaluation, node memories are updated batch by batch as events arrive in temporal order. The memory update function $f_{update}$ is implemented either as a GRU or RNN; we report results for both variants in the paper.
>
> **Event selection**: We use the MOSEK solver to select the events. We set the maximum solving time to three minutes, but in all our experiments the solver converged before this limit and returned a solution without early termination.
>
> **Pick the event set**: We do not use a separate optimizer to obtain the candidate event set. Instead, we obtain a candidate set by  memory backtracking tree and  topology attribution tree. In all experiments, the depth of the memory backtracking tree is fixed to 5.
>
> **random seeds**: The random seed is 42
>
> **code release**: We plan to release our code upon acceptance of the paper.

---

### Official Review · Reviewer_UNHa · 2025-10-26

**Soundness:** 3
**Presentation:** 3
**Contribution:** 3
**Rating:** 6
**Confidence:** 3

**Summary:**

This paper proposes an explanation method for attributing the output of temporal graph neural networks to the important historical events. Specifically, the proposed method first applies the LRP on the embedding module to compute the topology attribution tree, which estimates the node memories’ contribution to node embeddings. Then, the proposed method further propagates it to the historical events by applying LRP on the memory updating module. The important events are finally selected by solving an optimization problem based on the obtained event contribution matrix.

**Strengths:**

- The ideas of applying the two-step LRP to estimate the importance of historical events and formulating the selection problem as an optimization variant are natural.
- The paper is generally well-written and easy to follow
- The proposed method achieves better performance than the compared baselines.

**Weaknesses:**

- The proposed method appears computationally intensive, as it requires building a topology attribution/memory backtracking tree and solving an optimization problem for each prediction. The authors should provide a clear complexity analysis and compare it with baseline methods. Additionally, a runtime comparison is needed to demonstrate practical efficiency.
- Some parts of the paper need further clarification. 1)How is Equation (13) converted into Equation (14)? 2) How is the optimization of Equation (14) actually solved?
- Minor issues. 1) It is better to denote the shape of the matrices/vectors when they first appear. 2) The superscript $t$ of $\mathcal N_{u_k}^t$ in Line 159 should be $n$.

**Questions:**

Please see the Weaknesses.

---

> ### Author Response · Authors · 2025-11-26
> **Response to Reviewer UNHa**
>
> Thank you for taking the time to review our paper!
>
> >**Q1: The proposed method appears computationally intensive, as it requires building a topology attribution/memory backtracking tree and solving an optimization problem for each prediction.**
>
> **A1**: We provide the complexity analysis in **Appendix A.9**.
>
> **Construct the topology attribution tree**: if $f_{\textnormal{emb}}$ is the graph sum function, the complexity is $O\big(N \cdot d_m+2^L\cdot B\cdot n \cdot d_m)$. If the $f_{\textnormal{emb}}$ is the graph attention function, the complexity is $O\big(N \cdot d_m+2^L\cdot B\cdot (n+n^2) \cdot d_m)$, where $d_m$ is the dimension of the memory vector, $n$ is the number of most recent interactions considered, $L$ is the number of graph layers, $N = |\mathcal{V}(t)|$ is the number of nodes, and $B$ is the batch size.
>
> **Construct the memory backtracking tree**:  the complexity is $O\big(2^T\cdot T\cdot d_m\cdot d_m)$, where $T$ is the depth of memory backtracking tree, and $d_m$ is the dimension of the memory vector.
>
> **Select the important events**: the complexity is $O\big(d_c^3)$, where $d_c$ is the number of events in final contribution matrix $\mathbf{C}^t$
>
> Besides, in **Figure 7** in appendix, we report the running time with different depths T across all datasets. We partition the computation time into three components: **topology time**, for constructing the topology attribution tree and computing neighbor contributions; **memory time**, for constructing the memory backtracking tree and computing contributions of historical events; and **selection time**, for solving Eq. (14). Topology time is omitted for the enron, UCI, tgbn-trade, tgbn-genre and tgbn-reddit datasets because it accounts for less than 1% of the total runtime and is therefore not shown in the stacked bars. **For link prediction, memory time is the dominant cost**. It increases with the memory depth $T$ but remains practical: when $T = 10$, constructing the memory backtracking tree and computing historical contributions takes **at most 5 seconds**.  **For node property prediction, selection time dominates cost** because the output is higher-dimensional than in link prediction (where the logit is one-dimensional), making the final solving step more expensive. Even so, when $T = 10$, **event selection takes at most 3 seconds**, which is still acceptable.
>
> >**Q2: How is Equation (13) converted into Equation (14)?**
>
> **A2**: We show the derivation in **Appendix A.16**. In Eq. (14), we derived the KL divergence between $p_1$ and $p_2$, where $p_1$ and $p_2$ are arbitrary probabilities obtained by applying the sigmoid function. Now, we let $p_2$ be the predicted probability on the original temporal graph, $p_2=\hat{y}(\mathcal{E}(t))$, and $p_1$ be the predicted probability after **retaining only a subset of critical events and removing the others**. Based on the event-contribution matrix $\mathbf{C}^t$, we approximate the perturbed logits as $z_1=\sum_{l=1}^{d_c} d_l \mathbf{C}^{t}_{e_l}$, where $d_c$ is the number of candidate events, and $\mathbf{d} \in $ {0,1}$^{d_c}$ is a selection vector whose entry $d_l$ indicates whether event $e_l$ is selected.  Substituting $z\_1$ into Eq. (13) gives $\sigma(z_2) z_2- \hat{y}(\mathcal{E}(t)) \sum\_{l=1}^{d\_c} d\_l \mathbf{C}^{t}\_{e\_l}-sp(z_2)+sp(\sum\_{l=1}^{d\_c} d\_l \mathbf{C}^{t}\_{e\_l})$. Since $\sigma(z_2) z_2$ and $sp(z_2)$ do not depend on $\mathbf{d}$, they are constant with respect to the selection and can be dropped from the optimization objective. Minimizing the remaining KL term over $\mathbf{d}$ yields Eq. (14), which we use to select a small set of key events that best preserves the original prediction.
>
> >**Q3: How is the optimization of Equation (14) actually solved?**
>
> **A3**: We apologize for not explaining the optimization of Eq. (14) clearly and thank the reviewer for pointing this out. Eq. (14) can be solved in two ways: (1) **exact optimization**, where the selection variables are constrained to be binary (0/1) and the final explanatory events are directly determined by the variables equal to 1; and (2) **relaxed optimization**, where the selection variables are allowed to lie in [0,1] and the final explanatory events are obtained by taking the top-K variables with the largest values. We implement Eq. (14) in Python using the **CVXPY** library with **MOSEK** as the solver. All results reported in the paper are based on the **exact (binary) formulation**. In  **Figure 7** in appendix, we show the selection time for solving Eq. (14) across memory depths T and datasets. When $T = 10$, event selection takes at most 3 seconds, which is still acceptable.
>
> >**Q4: Minor issues. It is better to denote the shape of the matrices/vectors when they first appear. The superscript in Line 159 should be n.**
>
> **A4:** We thank the reviewer for pointing out these issues. We have revised them in the new version.

---

### Official Review · Reviewer_TxVo · 2025-10-27

**Soundness:** 3
**Presentation:** 3
**Contribution:** 3
**Rating:** 4
**Confidence:** 3

**Summary:**

This paper addresses the challenge of explainability in Temporal Graph Networks (TGNs). It proposes a novel framework called MemExplainer to attribute the predictions made by TGNs by considering both spatial and temporal dynamics. The framework introduces two components: the topology attribution tree, which attributes predictions based on the contributions of recent events and their associated node memories, and the memory backtracking tree, which traces how past events influence memory evolution. The method ensures that the total contribution from events equals the model’s output logits, maintaining a conservation property. Extensive experiments on seven temporal graph datasets demonstrate that the proposed method outperforms several explainability approaches, showing improved fidelity and lower sparsity in the explanations.

**Strengths:**

1. This work proposes an innovative framework for explaining TGNs. The dual-attribution approach—using a topology attribution tree and a memory backtracking tree—gives a more comprehensive explanation of model predictions by accounting for both spatial and temporal influences. The conservation principle ensures that the sum of all event contributions exactly matches the model’s output logits, which helps guarantee the explanation’s faithfulness.
2. The method shows strong performance across multiple benchmarks. It consistently outperforms existing explainers such as TGNNExplainer and TempME on both node property prediction and link prediction tasks.
3. The paper is supported by solid theoretical foundations and practical design. For example, building both the spatial and temporal attribution components on Layer-wise Relevance Propagation (LRP) provides a consistent and theoretically justified framework, enhancing both reliability and interpretability.

**Weaknesses:**

1. The paper's exploration of explainability for memory-based dynamic graph methods is commendable. However, it is questionable whether such methods can provide meaningful explanations on datasets with very high repetition rates, such as Wikipedia, Reddit, etc. This issue has been noted in recent research. For example, [1] points out a key limitation of these datasets: models can achieve good performance simply by remembering whether an edge has appeared before, making the learning task too easy. The strong performance of the EdgeBank model from [3]—which requires no training—on these datasets further supports this concern. Similarly, [2] analyzes attention weights in Transformer-based models and finds that models mainly rely on high-frequency, repetitive edges. This makes the explanations straightforward and intuitively clear, but also suggests they may not capture deeper reasoning for datasets with low repetition rate. Therefore, when evaluating dynamic graph models, using event repetition rate as a criterion for dataset selection might lead to more meaningful research.
2. Although the method performs well on several datasets, its computational cost—especially from the memory backtracking process—could be an issue for large graphs with long temporal histories. The approach involves recursive backtracking and memory updates, which may become slow on very large-scale data. In addition, while the optimization for selecting important events is defined in Equation 14, the paper lacks details on how this optimization is implemented efficiently in practice, especially as the number of events increases. More information about convergence behavior, runtime, and computational trade-offs would help readers assess its practicality.
3. While the framework is technically sound, the resulting explanations may not be easy to interpret. The tree structure (topology and memory) could be hard to understand without more explanation or visual examples. The paper would benefit from including concrete case studies or qualitative examples to show how specific predictions (on both low/high repetition rate) are explained and how users might apply these explanations in real-world scenarios.

Reference

[1] TGB-Seq Benchmark: Challenging Temporal GNNs with Complex Sequential Dynamics https://arxiv.org/abs/2502.02975v3

[2] TIDFormer: Exploiting Temporal and Interactive Dynamics Makes A Great Dynamic Graph Transformer https://arxiv.org/abs/2506.00431

[3] Towards Better Evaluation for Dynamic Link Prediction https://arxiv.org/abs/2207.10128

**Questions:**

Please see weaknesses.

---

> ### Author Response · Authors · 2025-11-26
> **Response to Reviewer TxVo (1)**
>
> Thank you for taking the time to review our paper!
>
> >**Q1: However, it is questionable whether such methods can provide meaningful explanations on datasets with very high repetition rates, such as Wikipedia, Reddit, etc. This issue has been noted in recent research.**
>
> **A1**: We think this is a very good question. We thank the reviewer for this insightful comment. We analyzed the important edges selected according to our method. For each target event, we use our method to obtain a set of important explanatory events $\mathcal{E}^\*$.  For each event $e=(u,v,t) \in \mathcal{E}^\*$, we compute its repetition count $r(e)$ defined as the total number of times the node pair (u, v) has appeared in the dynamic graph before the current prediction time. We then calculate the average repetition of the explanatory set as $\bar{r}=\frac{\sum_{e \in \mathcal{E}^\*} r(e)}{ |\mathcal{E}^\*|}$.
>
> On the Wikipedia, UCI, and Reddit datasets, we sample 500 target events and compute $\bar{r}$. The corresponding results are shown in **Figure 14**.  On the **UCI** dataset, about **50\%** of target events have $\bar{r}$ between **1 and 5.75**. On the **Wikipedia** dataset, about **28\%** of target events have $\bar{r}$ between **1 and 10**. On the **Reddit** dataset, about **60\%** of target events have $\bar{r}$ between **1 and 3**. Thus, on Wikipedia, important events often correspond to edges with high repetition counts, whereas on Reddit many important events involve edges that are rarely repeated.
>
> Besides, even when the underlying dataset has a high overall repetition rate, **the explanation task for a memory-based model is not reduced to merely checking whether an edge has appeared before**. In TGN, each event $e=(u,v,t)$ is associated with a **time-dependent memory state and temporal features**, so different repetitions of the same node pair can carry very different information. Our method therefore identify which specific **past occurrences and timestamps** of $(u,v)$ contribute most to the current prediction, not just whether $(u,v)$ is frequent. In TGNs, **explanations are not defined merely at the level of an edge (u,v); they must also specify when each interaction occurred**. As a result, even in highly repetitive graphs, selecting a sparse explanatory subset from a long history of interactions remains non-trivial, and the search space for explanations does not automatically shrink. Empirically, **Figure 14** shows that our method **does not simply focus on the most frequent edges, but also highlights events with relatively low repetition, especially on Reddit**.
>
> >**Q2: Although the method performs well on several datasets, its computational cost—especially from the memory backtracking process—could be an issue for large graphs with long temporal histories.**
>
> **A2**: We provide the complexity analysis in **Appendix A.9**.
>
> **Construct the topology attribution tree**: if $f_{\textnormal{emb}}$ is the graph sum function, the complexity is $O\big(N \cdot d_m+2^L\cdot B\cdot n \cdot d_m)$. If the $f_{\textnormal{emb}}$ is the graph attention function, the complexity is $O\big(N \cdot d_m+2^L\cdot B\cdot (n+n^2) \cdot d_m)$, where $d_m$ is the dimension of the memory vector, $n$ is the number of most recent interactions considered, $L$ is the number of graph layers, $N = |\mathcal{V}(t)|$ is the number of nodes, and $B$ is the batch size.
>
> **Construct the memory backtracking tree**:  the complexity is $O\big(2^T\cdot T\cdot d_m\cdot d_m)$, where $T$ is the depth of memory backtracking tree, and $d_m$ is the dimension of the memory vector.
>
> **Select the important events**: the complexity is $O\big(d_c^3)$, where $d_c$ is the number of events in final contribution matrix $\mathbf{C}^t$
>
> Besides, in **Figure 7** in appendix, we report the running time with different depths T across all datasets. We partition the computation time into three components: **topology time**, for constructing the topology attribution tree and computing neighbor contributions; **memory time**, for constructing the memory backtracking tree and computing contributions of historical events; and **selection time**, for solving Eq. (14). Topology time is omitted for the enron, UCI, tgbn-trade, tgbn-genre and tgbn-reddit datasets because it accounts for less than 1% of the total runtime and is therefore not shown in the stacked bars. **For link prediction, memory time is the dominant cost**. It increases with the memory depth $T$ but remains practical: when $T = 10$, constructing the memory backtracking tree and computing historical contributions takes **at most 5 seconds**.  **For node property prediction, selection time dominates cost** because the output is higher-dimensional than in link prediction (where the logit is one-dimensional), making the final solving step more expensive. Even so, when $T = 10$, **event selection takes at most 3 seconds**, which is still acceptable.

---

> ### Author Response · Authors · 2025-11-26
> **Response to Reviewer TxVo (2)**
>
> >**Q3: In addition, while the optimization for selecting important events is defined in Equation 14, the paper lacks details on how this optimization is implemented efficiently in practice, especially as the number of events increases.**
>
> **A3**: We apologize for not explaining the optimization of Eq. (14) clearly and thank the reviewer for pointing this out. Eq. (14) can be solved in two ways: (1) **exact optimization**, where the selection variables are constrained to be binary (0/1) and the final explanatory events are directly determined by the variables equal to 1; and (2) **relaxed optimization**, where the selection variables are allowed to lie in [0,1] and the final explanatory events are obtained by taking the top-K variables with the largest values. We implement Eq. (14) in Python using the **CVXPY** library with **MOSEK** as the solver. All results reported in the paper are based on the **exact (binary) formulation**. In  **Figure 7** in appendix, we show the selection time for solving Eq. (14) across memory depths T and datasets. When $T = 10$, event selection takes **at most 3 seconds**, which is still acceptable.
>
> >**Q4: The resulting explanations may not be easy to interpret. The tree structure (topology and memory) could be hard to understand without more explanation or visual examples.**
>
> **A4**: We would like to clarify that the final explanation provided by our framework **is not the tree structure itself**, but a sparse set of **important events**. The tree structure over topology and memory is only an internal device used to propagate and aggregate contribution scores for candidate events.
>
> We further evaluate our method on a **human pose estimation** task and present case studies that illustrate the interpretability of the resulting explanations. For each video, we first apply YOLOPose to detect human keypoints and construct a skeleton graph whose nodes are body joints and whose edges follow the human kinematic structure. This yields a sequence of skeleton graphs (one per frame), which we feed into TGN to predict the action label. We use the HMDB51 dataset, and select four intuitive classes, sit-up, pull-up, climbing, and running, as our training and evaluation subset.
>
> For each video, we run our method and all baselines to extract important edges and evaluate the resulting explanations. In addition to $Fidelity_{prob}$ and $Fidelity_{kl}$, we introduce $Fidelity_{label}$, which is set to 1 if the label predicted by TGN **using only the selected important edges** matches the true label, and 0 otherwise; we then average this value over all videos. **Figure 6 in appendix** reports the performance of $Fidelity_{prob}$,  $Fidelity_{kl}$, and $Fidelity_{label}$. **Higher** $Fidelity_{label}$ and **lower** $Fidelity_{prob}$ and $Fidelity_{kl}$ indicate better explanations. As shown in Figure 6, **our method consistently outperforms all baselines on this pose estimation task**.
>
> In **Tables 2, 3, 4, and 5**, we present case studies for the **pull-up**, **running**, **climbing**, and **sit-up** classes. Green nodes denote joint (bone) nodes, red edges denote the original skeletal connections, yellow edges denote the selected important edges, and each image corresponds to one frame from the video.
>
> Our method yields **sparse, time-aware** explanations that can be traced across representative frames of the sequence and are aligned with human biomechanical intuition. For **pull-ups**, the yellow edges concentrate on the **main kinematic chains** (e.g., shoulder–elbow–wrist and hip–knee) during both the lifting and lowering phases. For **running**, they focus on the **lower-limb chain (hip–knee–ankle)** that drives forward motion, making it clear when the discriminative gait pattern appears. For **climbing**, our method mainly selects the **supporting arm and leg** that hold the body and control the center of mass, rather than uniformly highlighting the entire skeleton.
> For **sit-ups**, the selected edges follow the **torso–hip–knee** chain, capturing how the upper body bends around relatively fixed legs.
>
> In contrast, across all four actions, **PGExplainer**, **GNNExplainer**, and **TGNNExplainer** typically produce **dense masks on a single frame, marking most skeletal links as important** and making it difficult to identify which joints actually drive the prediction. **TempME** often selects edges that fluctuate over time and are misaligned with the key biomechanics of the movement: in many frames, it highlights secondary body parts, and **its selected edges do not form a consistent pattern across frames**.

---

> > ### Comment · Reviewer_TxVo · 2025-11-27
> >
> > Thank you very much for the authors’ thoughtful response, the additional experiments, and the revised version of the manuscript! The supplementary analysis—particularly the investigation into how your model attends to important edges in different datasets with high repetition rates—was insightful and has largely addressed my initial concerns. However, I still think that it could be more compelling to see similar analyses conducted on datasets that inherently exhibit low repetition rates. In my view, such experiments could offer greater practical relevance by moving beyond the constraints of highly repetitive data.
> >
> > Additionally, I found the added human pose estimation experiments quite interesting. The visualizations clearly illustrate the differences among baselines in identifying structurally important edges of the human body. However, I do have some questions regarding Tables 2–5: why do the different baselines use varying numbers of images for the corresponding frames? Moreover, it appears that the images aligned vertically across these tables do not correspond to the same original frame. I believe that maintaining consistent settings across all methods in the visualization of the case studies—would further strengthen the persuasiveness of the results.

---

> > > ### Author Response · Authors · 2025-11-27
> > > **Response to Reviewer TxVo**
> > >
> > > We thank TxVo for the helpful reply.
> > > >**Q1: However, I still think that it could be more compelling to see similar analyses conducted on datasets that inherently exhibit low repetition rates. In my view, such experiments could offer greater practical relevance by moving beyond the constraints of highly repetitive data.**
> > >
> > > **A1**: We appreciate this suggestion and **agree** that evaluating our method on datasets with inherently low repetition would be highly informative. In this work, we followed prior studies on temporal graph learning and explanation by using standard benchmarks such as Wikipedia and Reddit, which are widely adopted and thus enable fair comparison. Importantly, our method does not assume high repetition and can in principle be applied to any temporal graph. A systematic evaluation on low-repetition datasets is an interesting extension, and we regard this as an important direction for future work.
> > >
> > > >**Q2: However, I do have some questions regarding Tables 2–5: why do the different baselines use varying numbers of images for the corresponding frames? Moreover, it appears that the images aligned vertically across these tables do not correspond to the same original frame**
> > >
> > > **A2**: We apologize for the confusion caused by our original description.
> > >
> > > First, the explanations are defined at the level of the **entire video sequence** rather than individual frames. For each frame in the video, we extract a human skeleton graph and feed these graphs into the Temporal Graph Network (TGN) in temporal order. The TGN then predicts the label based on the skeleton graphs from the whole input sequence.
> > >
> > > Second, for a fair comparison, all methods operate on exactly **the same input sequence of skeleton graphs for each video**. We also fix a **global edge budget**, i.e., the **total number of key human-pose edges** that each method is allowed to select for that video. The resulting explanations are visualized in Tables 2, 3, 4, and 5. **Different methods select different explanation edges, the explanation edges naturally fall on different frames and involve different bone connections**. Consequently, these explanations results are **not meant to be aligned frame-by-frame across methods**.
> > >
> > > Third, methods such as GNNExplainer, PGExplainer, and TGNNExplainer often mark **almost all skeleton edges** within a **single frame** as important. Under a fixed edge budget, **this behavior naturally yields fewer selected (visualized) frames**. In contrast, our method and TempME typically select **a much smaller subset of edges per frame**, which leads to more selected (visualized) frames under the same total edge budget.
> > >
> > > Finally, Tables 2–5 show that our method produces sparse, time-aware explanations that can be tracked across representative frames of the sequence and that align with human biomechanical intuition. By comparison, PGExplainer, GNNExplainer, and TGNNExplainer usually generate dense masks on a single frame, making it difficult to see which joints actually drive the prediction, while the edges selected by TempME do not form a consistent pattern across time.

---

### Official Review · Reviewer_Y5yW · 2025-11-02

**Soundness:** 3
**Presentation:** 3
**Contribution:** 3
**Rating:** 4
**Confidence:** 4

**Summary:**

The paper proposes MemExplainer, an explainability framework for Temporal Graph Networks (TGNs). It decomposes the model's prediction into contributions from historical interaction events through a two-stage attribution process: the Topology Attribution Tree and the Memory Backtracking Tree.

**Strengths:**

Timely and meaningful problem: Explaining TGNs is crucial in domains such as finance, recommendation, and fraud detection; focusing on memory influence is insightful.

Comprehensive method: The topology + memory two-step structure is well-described with algorithms and clear relevance propagation logic.

**Weaknesses:**

Unclear optimization in Eq. (14): The binary event-selection problem lacks details—whether solved exactly, heuristically, or via continuous relaxation—and its time complexity. Clarify the algorithm and provide runtime statistics.

Scalability and resource cost: Memory backtracking may explode for large event histories. The paper lacks complexity or runtime/memory analysis. Please add empirical resource tables or propose pruning strategies.

Baseline implementation details: How were static explainers (GNNExplainer, PGExplainer) adapted to TGNs? What hyperparameters and seeds were used for TGNNExplainer and TempME？

Robustness / statistical significance: Report averages ± std or confidence intervals for repeated runs, since some improvements are small.

Model-component assumptions: The derivations assume a GRU updater. Discuss applicability to other TGNs (e.g., attention-based or transformer-style updaters) and whether the LRP rules transfer directly.

Limited human-interpretability evaluation: Fidelity metrics show quantitative preservation, but user-level interpretability is not assessed.

**Questions:**

How exactly is Eq. (14) optimized? What algorithm and complexity are used, and what are the runtime/memory statistics across datasets?

Please provide empirical runtime and memory data for memory-backtracking with different depths L.

Can the approach generalize to non-GRU TGNs (e.g., attention updaters)? What modifications are required?

---

> ### Author Response · Authors · 2025-11-26
> **Response to Reviewer Y5yW (1)**
>
> Thank you for taking the time to review our paper!
>
> >**Q1: Unclear optimization in Eq. (14). How exactly is Eq. (14) optimized? What algorithm and complexity are used. what are the runtime/memory statistics across datasets?**
>
> **A1**: We apologize for not explaining the optimization of Eq. (14) clearly and thank the reviewer for pointing this out. Eq. (14) can be solved in two ways: (1) **exact optimization**, where the selection variables are constrained to be binary (0/1) and the final explanatory events are directly determined by the variables equal to 1; and (2) **relaxed optimization**, where the selection variables are allowed to lie in [0,1] and the final explanatory events are obtained by taking the top-K variables with the largest values. We implement Eq. (14) in Python using the **CVXPY** library with **MOSEK** as the solver. All results reported in the paper are based on the **exact (binary) formulation**. In  **Figure 7** in appendix, we show the selection time for solving Eq. (14) across memory depths T and datasets. When $T = 10$, event selection takes at most 3 seconds, which is still acceptable.
>
> >**Q2: Please provide empirical runtime and memory data for memory-backtracking with different depths T.**
>
> **A2:** In **Figure 7** in appendix, we report the running time with different depths T across all datasets. We partition the computation time into three components: **topology time**, for constructing the topology attribution tree and computing neighbor contributions; **memory time**, for constructing the memory backtracking tree and computing contributions of historical events; and **selection time**, for solving Eq. (14). Topology time is omitted for the enron, UCI, tgbn-trade, tgbn-genre and tgbn-reddit datasets because it accounts for less than 1% of the total runtime and is therefore not shown in the stacked bars. **For link prediction, memory time is the dominant cost**. It increases with the memory depth $T$ but remains practical: when $T = 10$, constructing the memory backtracking tree and computing historical contributions takes **at most 5 seconds**.  **For node property prediction, selection time dominates cost** because the output is higher-dimensional than in link prediction (where the logit is one-dimensional), making the final solving step more expensive. Even so, when $T = 10$, **event selection takes at most 3 seconds**, which is still acceptable.
>
> >**Q3: The paper lacks complexity analysis**
>
> **A3**: We provide the complexity analysis in **Appendix A.9**.
>
> **Construct the topology attribution tree**: if $f_{\textnormal{emb}}$ is the graph sum function, the complexity is $O\big(N \cdot d_m+2^L\cdot B\cdot n \cdot d_m)$. If the $f_{\textnormal{emb}}$ is the graph attention function, the complexity is $O\big(N \cdot d_m+2^L\cdot B\cdot (n+n^2) \cdot d_m)$, where $d_m$ is the dimension of the memory vector, $n$ is the number of most recent interactions considered, $L$ is the number of graph layers, $N = |\mathcal{V}(t)|$ is the number of nodes, and $B$ is the batch size.
>
> **Construct the memory backtracking tree**:  the complexity is $O\big(2^T\cdot T\cdot d_m\cdot d_m)$, where $T$ is the depth of memory backtracking tree, and $d_m$ is the dimension of the memory vector.
>
> **Select the important events**: the complexity is $O\big(d_c^3)$, where $d_c$ is the number of events in final contribution matrix $\mathbf{C}^t$
>
> >**Q4: Scalability and resource cost: Memory backtracking may explode for large event histories. Please add empirical resource tables or propose pruning strategies.**
>
> **A4**:  In our experiments, we reduced the time complexity by limiting the depth of the memory backtracking tree. In the main text, we set this depth to 5. All experiments were run on a MacBook Pro with an Apple M2 Pro chip and 32 GB of memory.
>
> >**Q5:  Baseline implementation details: How were static explainers (GNNExplainer, PGExplainer) adapted to TGNs? What hyperparameters and seeds were used for TGNNExplainer and TempME?**
>
> **A5**: For the static explainers (GNNExplainer and PGExplainer), we follow the experimental settings of TGNNExplainer. In that work, the authors reimplement PGExplainer and GNNExplainer and adapt them to temporal graphs, where these explainers assign a weight to each event rather than each edge. The authors then train the static explainers using their objectives, keeping the memory vector fixed. For TGNNExplainer, we use 50 rollouts per target event. For TempME, we train for 200 epochs. For both TGNNExplainer and TempME, the random seed is set to 42.

---

> ### Author Response · Authors · 2025-11-26
> **Response to Reviewer Y5yW (2)**
>
> >**Q6: Robustness / statistical significance: Report averages ± std or confidence intervals for repeated runs, since some improvements are small.**
>
> **A6**: We add there ablation variants as baselines and update the results in **Figure 2** and **Figure 3**.
>
> **w/o memory** removes the memory backtracking tree and selects events solely based on the contributions of neighboring events.
>
> **w/o topology** removes the topology attribution tree and selects events solely based on the contributions of historical events.
>
> **w/o selection** does not use the objective function to select events; instead, it accumulates event contributions and chooses the final explanatory events using a top-k rule.
>
> **Tables 6 and 7 in the appendix** report the mean and standard deviation of $Fidelity_{KL}$ and $Fidelity_{prob}$ corresponding to **Figures 2 and 3**. We further performed a t-test between our method and the second-best baseline, and found statistically significant differences in **77%** of the cases for $Fidelity_{KL}$ and in **74%** of the cases for $Fidelity_{prob}$.
>
> >**Q7: Discuss applicability to other TGNs (e.g., attention-based or transformer-style updaters) and whether the LRP rules transfer directly. Can the approach generalize to non-GRU TGNs (e.g., attention updaters)? What modifications are required?**
>
> **A7**: More generally, our framework can be extended to other TGNs **as long as the TGN’s memory update function and embedding/message function can be decomposed using LRP**, i.e., LRP can be used to assign contribution scores to their inputs. In such cases, the same Topology Attribution Tree and Memory Backtracking Tree proposed in our work can be applied to propagate and aggregate contributions over neighboring and historical events, followed by the same selection procedure.
>
> For **attention-based** updaters, we extend the LRP allocation rule to the graph attention mechanism.We follow the LRP formulation for graph attention networks in prior work [1] to propagate relevance through attention weights and message transformations. After that, we reuse the same Topology Attribution Tree described in the paper to aggregate the contributions of neighboring events over time. The corresponding experimental results are reported in **Figures 8 and 9**, **where our method, MemExplainer consistently outperforms baseline explainers across all datasets and both evaluation metrics.**
>
> We also test the performance of **RNN** updaters, we similarly apply LRP to the RNN memory update function and then use the same Memory Backtracking Tree to trace relevance back to historical events. Figures **10 and 11** show that, under these RNN updaters, **our method, MemExplainer again achieves better performance than the baselines on all datasets and metrics**.
>
> Regarding transformer-style memory updaters, our current method does not directly extend to this setting. In this work, the TGN uses an RNN/GRU as the memory update function, and we have not considered transformer-based updates. To support a transformer memory updater, one would first need to use LRP to decompose the contributions within the transformer module, and then plug these relevance scores into our framework by applying the Memory Backtracking Tree to obtain historical contributions and perform the final selection.
>
> [1] Attnlrp: attention-aware layer-wise relevance propagation for transformers.

---

> ### Author Response · Authors · 2025-11-26
> **Response to Reviewer Y5yW (3)**
>
> >**Q8: Limited human-interpretability evaluation: Fidelity metrics show quantitative preservation, but user-level interpretability is not assessed.**
>
> **A8**: We further evaluate our method on a **human pose estimation** task and present case studies that illustrate the interpretability of the resulting explanations. For each video, we first apply YOLOPose to detect human keypoints and construct a skeleton graph whose nodes are body joints and whose edges follow the human kinematic structure. This yields a sequence of skeleton graphs (one per frame), which we feed into TGN to predict the action label. We use the HMDB51 dataset, and select four intuitive classes, sit-up, pull-up, climbing, and running, as our training and evaluation subset.
>
> For a fair comparison, all explanation methods operate on exactly the same input sequence of skeleton graphs for each video. We also fix a **global edge budget**, i.e., the total number of key human-pose edges that each explanation method is allowed to select for that video. In addition to $Fidelity_{prob}$ and $Fidelity_{kl}$, we introduce $Fidelity_{label}$, which is set to 1 if the label predicted by TGN **using only the selected important edges** matches the true label, and 0 otherwise; we then average this value over all videos. **Figure 6 in appendix** reports the performance of $Fidelity_{prob}$,  $Fidelity_{kl}$, and $Fidelity_{label}$. **Higher** $Fidelity_{label}$ and **lower** $Fidelity_{prob}$ and $Fidelity_{kl}$ indicate better explanations. As shown in Figure 6, **our method consistently outperforms all baselines on this pose estimation task**.
>
> In **Tables 2, 3, 4, and 5**, we present case studies for the **pull-up**, **running**, **climbing**, and **sit-up** classes. Green nodes denote joint (bone) nodes, red edges denote the original skeletal connections, yellow edges denote the selected important edges, and each image corresponds to one frame from the video. Because different methods select different explanation edges, the **explanation edges** naturally fall on **different frames** and involve different bone connections. Consequently, these explanation results are not meant to be aligned frame-by-frame across methods.
>
>
> Our method yields **sparse, time-aware** explanations that can be traced across representative frames of the sequence and are aligned with human biomechanical intuition. Our method typically select a **much smaller subset of edges** per frame, which leads to **more selected (visualized) frames** under the same total edge budget.  For **pull-ups**, the yellow edges concentrate on the **main kinematic chains** (e.g., shoulder–elbow–wrist and hip–knee) during both the lifting and lowering phases. For **running**, they focus on the **lower-limb chain (hip–knee–ankle)** that drives forward motion, making it clear when the discriminative gait pattern appears. For **climbing**, our method mainly selects the **supporting arm and leg** that hold the body and control the center of mass, rather than uniformly highlighting the entire skeleton.
> For **sit-ups**, the selected edges follow the **torso–hip–knee** chain, capturing how the upper body bends around relatively fixed legs.
>
> In contrast, across all four actions, **PGExplainer**, **GNNExplainer**, and **TGNNExplainer** typically produce **dense masks on a single frame, marking most skeletal links as important** and making it difficult to identify which joints actually drive the prediction.  Under a fixed edge budget, this behavior naturally yields **fewer selected (visualized)** frames. **TempME** often selects edges that fluctuate over time and are misaligned with the key biomechanics of the movement: in many frames, it highlights secondary body parts, and **its selected edges do not form a consistent pattern across frames**.

---

### Author Response · Authors · 2025-12-03
**The summary response to all reviewers**

**We summarize here the main concerns raised by the reviewers (Y5yW, TxVo, UNHa, 6Mxv) and how we addressed them in the rebuttal and appendix.**
>**Q1: How exactly is Eq. (14) optimized?**

**A1**: Four reviewers (Y5yW, TxVo, UNHa, and 6Mxv) asked this question. We implement Eq. (14) in Python using the **CVXPY** library with **MOSEK** as the solver. All results reported in the paper are based on the **exact (binary) formulation**. In exact formulation, the selection variables are constrained to be binary (0/1) and the final explanatory events are directly determined by the variables equal to 1. In  **Figure 7** in appendix, we show the selection time for solving Eq. (14) across memory depths T and datasets. When $T = 10$, event selection takes at most 3 seconds, which is still acceptable.

>**Q2:The paper lacks complexity analysis and the authors should provide empirical runtime on different datasets.**

**A2:** Four reviewers (Y5yW, TxVo, UNHa, and 6Mxv) asked this question. We provide the complexity analysis in **Appendix A.9** and  report the running time with different depths T across all datasets In **Figure 7** in appendix. We partition the computation time into three components: **topology time**, for constructing the topology attribution tree and computing neighbor contributions; **memory time**, for constructing the memory backtracking tree and computing contributions of historical events; and **selection time**, for solving Eq. (14).  **For link prediction, memory time is the dominant cost and remains practical**: when $T = 10$, it takes **at most 5 seconds**.  **For node property prediction, selection time dominates cost** because the output is higher-dimensional than in link prediction (where the logit is one-dimensional), making the final solving step more expensive. Even so, when $T = 10$, **event selection takes at most 3 seconds**, which is still acceptable.

>**Q3: Limited human-interpretability evaluation**

**A3:** Three reviewers (Y5yW, TxVo, and 6Mxv) asked this question. We further evaluate our method on a human pose estimation task and present case studies that illustrate the interpretability of the resulting explanations in **Appendix A.8**. As shown in Figure 6, our method consistently **outperforms** all baselines in three metrics on this pose estimation task. In Tables 2, 3, 4, and 5, we present case studies for the pull-up, running, climbing, and sit-up classes. Our method yields **sparse, time-aware** explanations that can be traced across representative frames of the sequence and are aligned with human biomechanical intuition.

>**Q4: Do the explanations transfer across different TGN variants?**

**A4:** Two reviewers (Y5yW, and 6Mxv) asked this question. More generally, our framework can be extended to other TGNs **as long as the TGN’s memory update function and embedding/message function can be decomposed using LRP**. In such cases, the same Topology Attribution Tree and Memory Backtracking Tree proposed in our work can be applied to propagate and aggregate contributions over neighboring and historical events, followed by the same selection procedure.

For **attention-based** updaters, we extend the LRP allocation rule to the graph attention mechanism. The corresponding experimental results are reported in **Figures 8 and 9**, **where our method, MemExplainer consistently outperforms baseline explainers across all datasets and both evaluation metrics.**

We also test the performance of **RNN** updaters. Figures **10 and 11** show that, under these RNN updaters, **our method, MemExplainer again achieves better performance than the baselines on all datasets and metrics**.

>**Q5: Statistics and significance testing are not reported.**

**A5:** Two reviewers (Y5yW, and 6Mxv) asked this question. **Tables 6 and 7 in the appendix** report the mean and standard deviation of $Fidelity_{KL}$ and $Fidelity_{prob}$ corresponding to **Figures 2 and 3**. We further performed a t-test between our method and the second-best baseline, and found statistically significant differences in **77%** of the cases for $Fidelity_{KL}$ and in **74%** of the cases for $Fidelity_{prob}$.

>**Q6: Key ablations are missing.**

**A6:** We include these three methods as baselines in **all experiments**.  **w/o memory** removes the memory backtracking tree and selects events only based on the contributions of neighboring events. **w/o topology** removes the topology attribution tree and selects events solely based on the contributions of historical events. **w/o selection** does not use the objective function to select events; instead, it accumulates event contributions and chooses the final explanatory events using a top-k rule. **The results show that our method outperforms baseline explainers across all experiments.**

**For reviewer-specific concerns and additional technical details, we respectfully refer the AC and reviewers to our point-by-point rebuttal.**

---

### Meta-Review · Area_Chair_xCr2 · 2026-01-04

**Summary:**

The paper proposes a novel explanation method for temporal graph networks, called MemExplainer. The approach relies on a two-stage LRP-based procedure to construct a topology attribution tree (spatial dependencies) and a memory backtracking tree (temporal contributions). Experiments on commonly used event-based temporal graph benchmarks aim to assess the method’s effectiveness using fidelity-based evaluation metrics.

The reviewers have initially raised several concerns spanning clarity (lack of details), reproducibility, scalability and computational costs, choice of benchmarks, lack of ablation studies, statistical significance, and human-level interpretability. While the rebuttal and discussion phase helped alleviate some of these concerns through clarifications and additional experimental results, several issues remain insufficiently addressed. In particular, I believe the paper would benefit from a more thorough evaluation of computational cost relative to existing baselines, the inclusion of harder benchmarks, and a clearer demonstration of the statistical significance of the reported improvements (over baselines).

It is also worth noting that only one of the four reviewers initially leaned toward acceptance. Even under somewhat optimistic predictions regarding score updates after the rebuttal, the overall acceptance support from the reviewers remains weak. Thus, I recommend rejection this time, while encouraging the authors to further strengthen the work by addressing the above issues in a future submission.

**Reviewer Concerns:**

**Concerns addressed by the rebuttal**

Several technical and methodological issues raised by reviewers were clarified during the rebuttal. In particular, authors clarified they employ the MOSEK solver (in CVXPY library) to the optimization procedure in Eq. (14) (Y5yW, TxVo, UNHa, and 6Mxv). Additional clarifications were also provided on how static explainers (e.g., GNNExplainer) were adapted to TGNs (``Y5yW``), following prior work.

Overall, the rebuttal further strengthened the empirical evaluation by:
- adding experiments addressing limited applicability assumptions, such as the reliance on GRU-based memory modules (Y5yW, 6Mxv)
- including additional experiment on human pose estimation to assessing alignment with user-level interpretability (Y5yW, TxVo, and 6Mxv).

Finally, concerns regarding baselines were partially mitigated through clearer positioning of TGNNExplainer and TempME as the main competitors (6Mxv).


**Concerns partially or not addressed**

A recurring concern across reviewers relates to computational cost and scalability (Y5yW, TxVo, UNHa, 6Mxv). While the authors now report time complexity (without providing details), critical steps are exponential (e.g., in the memory horizon) or cubic (e.g., in the number of events). Also, while the authors report empirical runtime for different datasets (in Tables 6 and 7), no empirical  comparisons against baselines are provided. Overall, I found difficult to assess the practical feasibility of the approach in real large-scale scenarios.

Significance of the results was also mentioned throughout the reviews (by Y5yW, TxVo, and 6Mxv) due to the lack of error bars. Although Tables 6 and 7 now include standard deviations and results from t-test, key details are missing (e.g., significance level and number of repetitions). Moreover, reported gains over baselines (e.g., TempME) appear marginal, standing less than one standard deviation difference, limiting the significance of the results.

Finally, some important concerns remain unaddressed. The choice of benchmarks was questioned by Reviewer TxVo in light of the known repetition rate issue. However, this was not addressed by the rebuttal — fact that was also acknowledged by the reviewer in their last reply.

**Reviewer Scores:**

Reviewer Y5yW raised several concerns that were addressed during the rebuttal, including limited applicability, user-level interpretability, and clarifications regarding the adaptation of static explainers. However, other issues, most notably scalability and resource cost, remain not fully addressed. Overall, I believe these clarifications could reasonably justify an improved assessment, potentially moving the score from 4 to 6.

Reviewer TxVo assigned a score of 4 (weak reject) and raised a few relevant concerns, mainly related to choice of benchmarks, computational trade-offs, and interpretability. While the reviewer acknowledged that the authors’s answer has largely addressed their initial concerns, it seems that reliance on the initial choice of benchmarks may still keep them from recommending acceptance. As such, it seems plausible that the reviewer would either maintain the original assessment or revise it upward to a weak accept (score of 6).

Reviewer UNHa mainly questioned the computational cost of the proposed approach and asked for clarifications. I guess that reviewer would maintain their positive evaluation (rating 6). The lack of cost comparison with that of baselines and the scalability of the method remain valid concerns preventing a higher score.

Finally, Reviewer 6Mxv raised several concerns, some of which were partially addressed. While some clarifications were helpful, key issues related to scalability, missing ablation studies, and the statistical significance of the results persist. Thus, I believe this reviewer would still lean toward rejection, though the additional clarifications might justify an increase from 2 to 4.

---

### Decision · Program_Chairs · 2026-01-26

Reject